∂ | **Open Peer Review** | Virology | Research Article

# Integrase-associated niche differentiation of endogenous large DNA viruses in crustaceans

Satoshi Kawato,[1] Reiko Nozaki,[1] Hidehiro Kondo,[1] Ikuo Hirono[1]

**ABSTRACT**  Crustacean genomes harbor sequences originating from nimaviruses, a family of large double-stranded DNA viruses infecting crustaceans. In this study, we recovered metagenome-assembled genomes of 27 endogenous nimaviruses from crustacean genome data. Phylogenetic analysis revealed four major lineages within *Nimaviridae*, and for three of these lineages, we propose novel genera of endogenous nimaviruses: "Majanivirus" and "Pemonivirus" identified from penaeid shrimp genomes, and "Clopovirus" identified from terrestrial isopods. Majanivirus genomes contain multiple eukaryotic-like genes such as baculoviral inhibitor of apoptosis repeat-containing genes, innexins, and heat shock protein 70-like genes, some of which contain introns. An alignment of long reads revealed that each endogenous nimavirus species specifically inserts into host microsatellites or within 28S rDNA. This insertion preference was associated with the type of virus-encoded DNA recombination enzymes, the integrases. Majaniviruses, pemoniviruses, some whispoviruses, and possibly clopoviruses specifically insert into the arthropod telomere repeat motif (TAACC/GGTTA)n and all possessed a specific tyrosine recombinase family. Pasiphaea japonica whispovirus and Portunus trituberculatus whispovirus, the closest relatives of white spot syndrome virus, integrate into the host 28S rDNA and are equipped with members of another family of tyrosine recombinases that are distantly related to telomere-specific tyrosine recombinases. Endogenous nimavirus genomes identified from sesarmid crabs, which lack tyrosine recombinases and are flanked by a 46-bp inverted terminal repeat, integrate into (AT/TA)n microsatellites through the acquisition of a Ginger2-like cut-and-paste DDE transposase. These results suggest that endogenous nimaviruses are giant transposable elements that occupy different sequence niches through the acquisition of different integrase families.

**IMPORTANCE**  Crustacean genomes harbor sequences originating from a family of large DNA viruses called nimaviruses, but it is unclear why they are present. We show that endogenous nimaviruses selectively insert into repetitive sequences within the host genome, and this insertion specificity was correlated with different types of integrases, which are DNA recombination enzymes encoded by the nimaviruses themselves. This suggests that endogenous nimaviruses have colonized various genomic niches through the acquisition of integrases with different insertion specificities. Our results point to a novel survival strategy of endogenous large DNA viruses colonizing the host genomes. These findings may clarify the evolution and spread of nimaviruses in crustaceans and lead to measures to control and prevent the spread of pathogenic nimaviruses in aquaculture settings.

**KEYWORDS**  WSSV, *Nimaviridae*, endogenous viral elements, transposable elements, tyrosine recombinase, integrase

Address correspondence to Ikuo Hirono, hirono@kaiyodai.ac.jp.

The authors declare no conflict of interest.

See the funding table on p. 20.

10.1128/spectrum.00559-23 **1**

*Nimaviridae* is a family of double-stranded DNA viruses infecting crustaceans (1). The only officially recognized member, white spot syndrome virus (WSSV), is the most devastating viral pathogen affecting global shrimp aquaculture (1–4). Although several other crustacean viruses have been reported to exhibit morphological characteristics similar to those of nimaviruses (5, 6), only one virus, Chionoecetes opilio bacilliform virus (CoBV), has been verified at the sequence level (NCBI Accession no. BDLS01000001-BDLS01000002, LC741431).

Despite the limited number of known exogenous nimaviruses, genomic analyses of decapod crustaceans have revealed the presence of sequences originating from nimaviruses (7–12). These endogenous viral elements (12, 13) are present as multi-copy elements sometimes reaching hundreds of copies per haploid genome (7, 10). However, the biological significance of these endogenous nimaviruses is unknown, and they do not exhibit any virulence.

In this study, we reconstructed 24 complete genomes and three partial genomes of endogenous nimaviruses recovered from crustacean genome data. Our results indicate that these viruses preferentially integrate into specific motifs in the host genome and that this insertion specificity is tightly linked with the presence of different integrase-like enzymes encoded by the viral genomes. These observations suggest that endogenous nimaviruses are selfish genetic elements that have colonized the crustacean genomes.

## RESULTS

### Metagenome-assembled genomes of endogenous nimaviruses

Genome sequencing of 17 crustacean genomes yielded 19–25 gigabases (Gb) of Illumina reads per genome and 769 Mb to 16.8 Gb of ONT reads per genome (Table 1; see Table S1 for detailed sequencing statistics). We also analyzed publicly available sequence data of the swimming crab *Portunus trituberculatus* (14), the blue shrimp *Litopenaeus stylirostris* (NCBI SRA Accession no. SRR12476764), the pink shrimp *Farfantepenaeus duorarum* (15), and the terrestrial isopod *Trachelipus rathkii* (16) (Table S2).

Our analysis yielded a total of 27 endogenous nimaviral genomes, 24 of which were regarded as complete (Table 2). Of these, 23 genomes were deposited in the DDBJ/NCBI/ENA databases as metagenome-assembled genomes (MAGs) of uncultivated virus genomes. The genomes of Portunus trituberculatus whispovirus (PotrWSV), Litopenaeus stylirostris majanivirus (LsMJNV), Farfantepenaeus duorarum majanivirus (FdMJNV), and Trachelipus rathkii clopovirus (TrCLPV) are available as supplementary files of the manuscript (see "Data availability" for the link to the FigShare repository). These MAGs are consensus sequences of closely related clones infecting a single organism. Most of the coding sequences on the assembled genomes are intact, but the actual individual copies within the host genome may be disrupted by mutations. Despite these limitations, the MAGs of endogenous nimaviruses represent distinct lineages of nimaviruses and provide valuable information for analyzing the evolution of *Nimaviridae*.

We estimated the copy numbers of endogenous nimaviruses by calculating the genome sequencing coverage of the host from the volume of Illumina read data plus the estimated host genome sizes (14, 17–19). The estimated copy numbers per haploid genome ranged from 22 (PotrWSV) to 477 (Marsupenaeus japonicus endogenous nimavirus; MjeNMV; LC738868.1) (Table S3). The abundance of MjeNMV copies in the kuruma shrimp genome aligns with previous estimates (7, 20) .

Maximum phylogenetic analysis of nimaviral core proteins (10, 21) revealed four major clusters within *Nimaviridae* (Fig. 1). As discussed below, we believe that these clades represent distinct genus-level taxa.

### Majaniviruses colonize telomere repeats of penaeid shrimp genomes

We previously reported on a group of penaeid shrimp-specific endogenous nimaviruses, exemplified by MjeNMV (Fig. 2A) (10). We propose for these penaeid endogenous nimaviruses a genus-level cluster, "Majanivirus" (**Ma**rsupenaeus **ja**ponicus endogenous

**TABLE 1** Crustacean samples sequenced in this study

| Genus | Species | Isolate | Geographic loaction | Year | BioSample accession |
|---|---|---|---|---|---|
| *Marsupenaeus* | *japonicus* | Aichi2020 | Japan: Aichi | 2020 | SAMD00511450 |
| *Melicertus* | *latisulcatus* | Mellat | Japan: Okinawa, East China Sea | 2016 | SAMD00111282 |
| *Penaeus* | *monodon* | Penmon | Japan: Aichi, Mikawa Bay | 2016 | SAMD00111283 |
| *Penaeus* | *semisulcatus* | Kagawa2020 | Japan: Kagawa | 2020 | SAMD00511443 |
| *Litopenaeus* | *vannamei* | Litvan | Japan | 2015 | SAMD00111285 |
| *Metapenaeus* | *ensis* | Metens | Japan: Aichi, Mikawa Bay | 2016 | SAMD00111287 |
| *Metapenaeus* | *joyneri* | Tokushima2020 | Japan: Tokushima | 2020 | SAMD00511444 |
| *Metapenaeopsis* | *lamellata* | Hokkoku2021 | Japan | 2021 | SAMD00511449 |
| *Trachysalambria* | *curvirostris* | Ube2021 | Japan: Yamaguchi, Ube | 2021 | SAMD00511448 |
| *Sicyonia* | sp. | Fukuoka2019 | Japan: Fukuoka | 2019 | SAMD00513257 |
| *Pasiphaea* | *japonica* | Toyama2020 | Japan: Toyama, Toyama Bay | 2020 | SAMD00511445 |
| *Hemigrapsus* | *takanoi* | Hemtak | Japan: Tokyo, Tokyo Bay | 2016 | SAMD00111291 |
| *Orisarma (Sesarmops)* | *intermedium* | Sesint | Japan: Kochi | 2016 | SAMD00111292 |
| *Orisarma (Chiromantes)* | *dehaani* | Chideh | Japan: Kochi | 2016 | SAMD00111293 |
| *Armadillidium* | *vulgare* | TUMSAT20210906 | Japan: Tokyo | 2021 | SAMD00511446 |
| *Porcellio* | *scaber* | TUMSAT20211004 | Japan: Tokyo | 2021 | SAMD00511447 |

<u>ni</u>mavirus), consisting of 12 members (Table 2). Complete majaniviral genomes range from 278 to 401 kb in size, with GC content ranging from 27% to 42%. Majanivirus genomes were recovered from all penaeid shrimp genome data sequenced in this study, except for *Sicyonia* sp. Fukuoka2019. We also identified partial majaniviral genomes from publicly available Illumina genome shotgun sequencing data of two penaeid shrimp genomes, *Litopenaeus stylirostris* and *Farfantepenaeus duorarum*.

Majaniviruses identified from *Penaeus sensu lato* (*Penaeus s. l.*: *Marsupenaeus*, *Melicertus*, *Fenneropenaeus*, *Litopenaeus*, *Farfantepenaeus*, and *Penaeus sensu stricto*) form a coherent and exclusive clade, indicative of close association and host selectivity (Fig. 1; Fig. S1). However, the phylogeny of *Penaeus s. l.*-associated majaniviruses does not simply reflect that of the host; instead, they are divided into two geographically defined clusters: the Indo-Western Pacific (IWP) and Atlantic-Eastern Pacific (AEP) (Fig. S1). This means that, in addition to the phylogeny of the host species, their geographic distribution has also influenced the diversification of majaniviruses.

We previously showed that the MjeNMV genome is chromosomally integrated into the kuruma shrimp genome (10), but their integration sites remained unknown. Bao et al. (11) were the first to show that Nimav-1_LVa (LC738872.1), a majanivirus, specifically insert into the (TAACC/GGTTA)n motifs in the genome of the Pacific white shrimp *Litopenaeus vannamei* (11). Our analysis of ONT read alignments indicates that MjeNMV and other complete majanivirus genomes are flanked by the same (TAACC/GGTTA)n pentanucleotide motifs (Fig. 2B), strongly suggesting that telomere insertion is a common feature of majaniviruses. However, some ONT reads were successfully mapped from one end of the majanivirus genome, spanning across the external (TAACC/GGTTA)n tract and reaching the other end of the genome. This suggests that some majaniviral copies could exist as concatemers, episomes, or possibly as a combination of both. This suggests that they are, or were until recently, actively replicating within the host genome.

A salient feature of majanivirus genomes is the expansion of eukaryotic-like genes (Fig. 3). The earliest reports on WSSV-like sequences in the penaeid shrimp genomes noted an expansion of a large DNA segment containing WSSV homologs as well as various eukaryotic genes, including baculoviral inhibitor of apoptosis repeat (BIR)-containing proteins (7) and an HSP70 homolog (20). Bao et al. also observed the presence of eukaryotic-like genes on the Nimav-1_LVa genome (11). The availability of complete majaniviral genomes confirms that the presence of eukaryotic-like genes is a shared trait of majaniviruses. Heat shock protein 70-like proteins (MjHSP70-2) (20) and innexins form their own clades on the phylogenetic trees, indicating that they have been vertically

**TABLE 2** Nimaviral genomes characterized in this study

| Proposed genus | Species | Host | Abbreviation | Accession | Length | GC% | CDS | Completeness | Coverage | |
| --- | --- | --- | --- | --- | --- | --- | --- | --- | Illumina | ONT |
| "Majanivirus" | Marsupenaeus japonicus endogenous nimavirus | Marsupenaeus japonicus | MjeNMV | LC738868.1 | 306,008 | 33 | 111 | Complete | 23,466.0 | 615.1 |
| | Melicertus latisulcatus majanivirus | Melicertus latisulcatus | MelaMJNV | LC738874.1 | 287,061 | 32 | 104 | Complete | 769.5 | 173.7 |
| | Metapenaeopsis lamellata majanivirus | Metapenaeopsis lamellata | MellatMJNV | AP027153.1 | 267951 | 27 | 106 | Complete | 118.9 | 6.6 |
| | Penaeus monodon majanivirus A | Penaeus monodon | PemoMJNVA | LC738870.1 | 294144 | 40 | 115 | Complete | 649.4 | 69.4 |
| | Penaeus monodon majanivirus B | Penaeus monodon | PemoMJNVB | LC738871.1 | 360,489 | 35 | 114 | Complete | 325.9 | 22.4 |
| | Litopenaeus vannamei majanivirus Nimav-Lv_1 | Litopenaeus vannamei | LvMJNV | LC738872.1 | 280452 | 35 | 119 | Complete | 116.7 | 8.3 |
| | Penaeus semisulcatus majanivirus | Penaeus semisulcatus | PeseMJNV | LC738873.1 | 291934 | 42 | 110 | Complete | 469.4 | 45.2 |
| | Metapenaeus ensis majanivirus | Metapenaeus ensis | MeenMJNV | LC738876.1 | 292,272 | 29 | 101 | Complete | 207.1 | 17.0 |
| | Metapenaeus joyneri majanivirus | Metapenaeus joyneri | MejoMJNV | LC738878.1 | 401182 | 26 | 124 | Complete | 169.8 | 17.9 |
| | Trachysalambria curvirostris majanivirus | Trachysalambria curvirostris | TrcuMJNV | LC738879.1 | 283150 | 28 | 101 | Complete | 134.2 | 14.7 |
| | Litopenaeus stylirostris majanivirus | Litopenaeus stylirostris | LsMJNV | NA[a] | 198899 | 36.03 | 77 | Partial | 100.2 | NA |
| | Farfantepenaeus duorarum majanivirus | Farfantepenaeus duorarum | FdMJNV | NA[a] | 217280 | 41.21 | 80 | Partial | 1,802.4 | NA |
| "Pemonivirus" | Marsupenaeus japonicus pemonivirus | Marsupenaeus japonicus | MjPMNV | AP027202.1 | 323944 | 48 | 102 | Complete | 1,187.1 | 4.7 |
| | Melicertus latisulcatus pemonivirus | Melicertus latisulcatus | MelaPMNV | LC738875.1 | 359647 | 48 | 109 | Complete | 212.6 | 41.6 |
| | Penaeus monodon endogenous nimavirus | Penaeus monodon | PmeNMV | LC738869.1 | 300002 | 45 | 94 | Complete | 892.6 | 95.2 |
| | Penaeus semisulcatus pemonivirus | Penaeus semisulcatus | PesePMNV | AP027152.1 | 277334 | 43 | 102 | Complete | 157.0 | 6.5 |
| Whispovirus | Hemigrapsus takanoi nimavirus | Hemigrapsus takanoi | HtNMV | LC738882.1 | 251731 | 47 | 111 | Partial | 383.7 | 19.4 |
| | Trachysalambria curvirostris nimavirus | Trachysalambria curvirostris | TrcuNMV | LC738880.1 | 331684 | 47 | 107 | Complete | 706.6 | 41.8 |
| | Sesarmops intermedium nimavirus | Sesarmops intermedium | SiNMV | LC738884.1 | 267936 | 44 | 104 | Complete | 79.1 | 50.2 |
| | Chiromantes dehaani nimavirus | Chiromantes dehaani | CdNMV | AP027155.1 | 285096 | 44 | 99 | Complete | 53.7 | 30.2 |
| | Metapenaeus ensis nimavirus | Metapenaeus ensis | MeNMV | LC738877.1 | 341,283 | 44 | 117 | Complete | 517.7 | 55.1 |
| | Sicyonia whispovirus | Sicyonia sp. | SicyWSV | LC738881.1 | 347,493 | 54 | 89 | Complete | 1351.1 | 237.7 |
| | Portunus trituberculatus whispovirus | Portunus trituberculatus | PotrWSV | NA[a] | 323740 | 48 | 98 | Complete | 102.1 | 72.5 |
| | Pasiphaea japonica whispovirus | Pasiphaea japonica | PajaWSV | LC738885.1 | 276272 | 35 | 86 | Complete | 159.4 | 32.4 |
| "Clopovirus" | Armadillidium vulgare clopovirus | Armadillidium vulgare | AvCLPV | LC738883.1 | 416,069 | 32 | 120 | Complete | 315.6 | 78.0 |
| | Porcellio scaber clopovirus | Porcellio scaber | PsCLPV | AP027154.1 | 509411 | 31 | 179 | Complete | 1224.2 | 99.5 |
| | Trachelipus rathkii clopovirus | Trachelipus rathkii | TrCLPV | NA[a] | 579413 | 35 | 196 | Complete | 125.4 | 112.8 |

[a]Not applicable to third-party annotation.

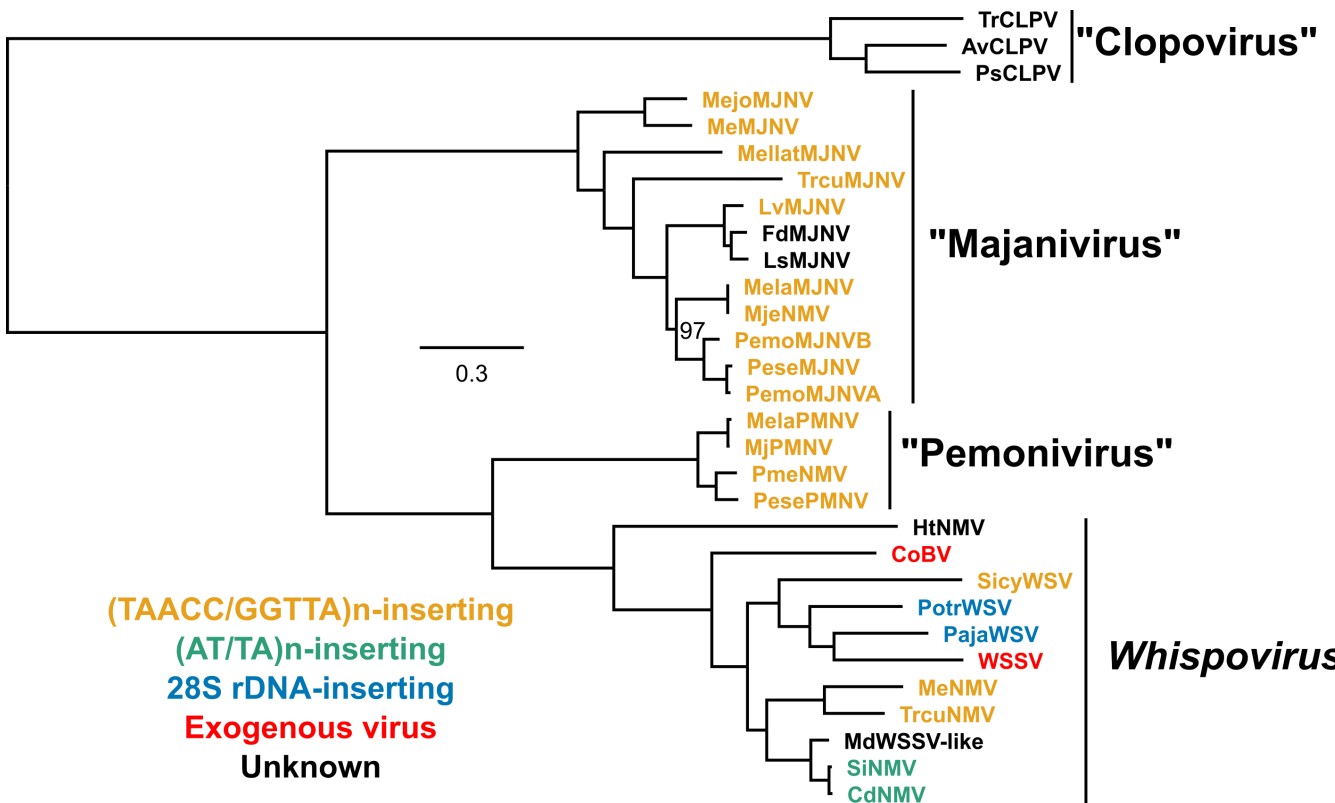

**FIG 1** Phylogenomic tree of *Nimaviridae*. Amino acid sequences of nine nimaviral core genes (wsv026, wsv282, wsv289, wsv303, wsv343, wsv360, wsv433, wsv447, and wsv514; 12,905 amino acids; substitution model: JTT + F + I + I + R5) were used for the analysis. Virus names are colored according to insertion motif specificity as indicated on the lower left. The bar in the middle of the figure denotes substitution per site. Ultrafast bootstrap value (1,000 trials) was 100% unless indicated beside the node. Proposed genus names are quoted and unitalicized. WSSV, white spot syndrome virus; CoBV, *Chionoecetes opilio* bacilliform virus; MdWSSV-like, *Metopaulias depressus* WSSV-like virus; see Table 2 for the abbreviations for the other viruses.

inherited from a common ancestor of the majaniviruses (Fig. 3B and C). BIR-containing proteins clustered with other decapod proteins, but we surmise that they are nimaviral sequences annotated as host genes (Fig. 3A). These findings demonstrate that majaniviruses harbor multiple eukaryotic-like genes, which were likely acquired from their decapod hosts.

Nimaviral core genes are a set of genes that are ubiquitously conserved among *Nimaviridae* and are likely to play essential functions in the viral replication cycle (10, 11, 21). The original nimaviral core gene set consisted of 28 genes. Bao et al. proposed the inclusion of four additional genes (wsv112, wsv206, wsv226, and wsw308) to this set, raising the total number to 32 (11). Two protein-coding genes lying downstream of the wsv306-like protein gene in the majaniviral genomes were suspected to be wsv308 and wsv310 orthologs, but their orthology could not be verified by sequence similarity due to substantial divergence. Regardless, structural prediction with ColabFold (22, 23) yielded remarkably similar predicted structures, with DALI Z-scores of 19.7 for wsv308-like proteins and 10.5 for wsv310-like proteins (Fig. S2 and S3; Files S1 and S2) (24). We, therefore, concluded that these two genes are authentic WSSV orthologs and added wsv308 and wsv310 to the nimaviral core gene repertoire. Our analysis supported the inclusion of wsv226, wsv308, and wsv310 to the core genes, but phylogenetic analysis of wsv112 and wsv206 suggested that they were acquired independently in the majaniviruses and whispoviruses (Fig. S4), although this does not necessarily mean that their functions are dispensable for viral replication. Consequently, our revised version of the nimaviral core gene set includes 31 genes (Table 3).

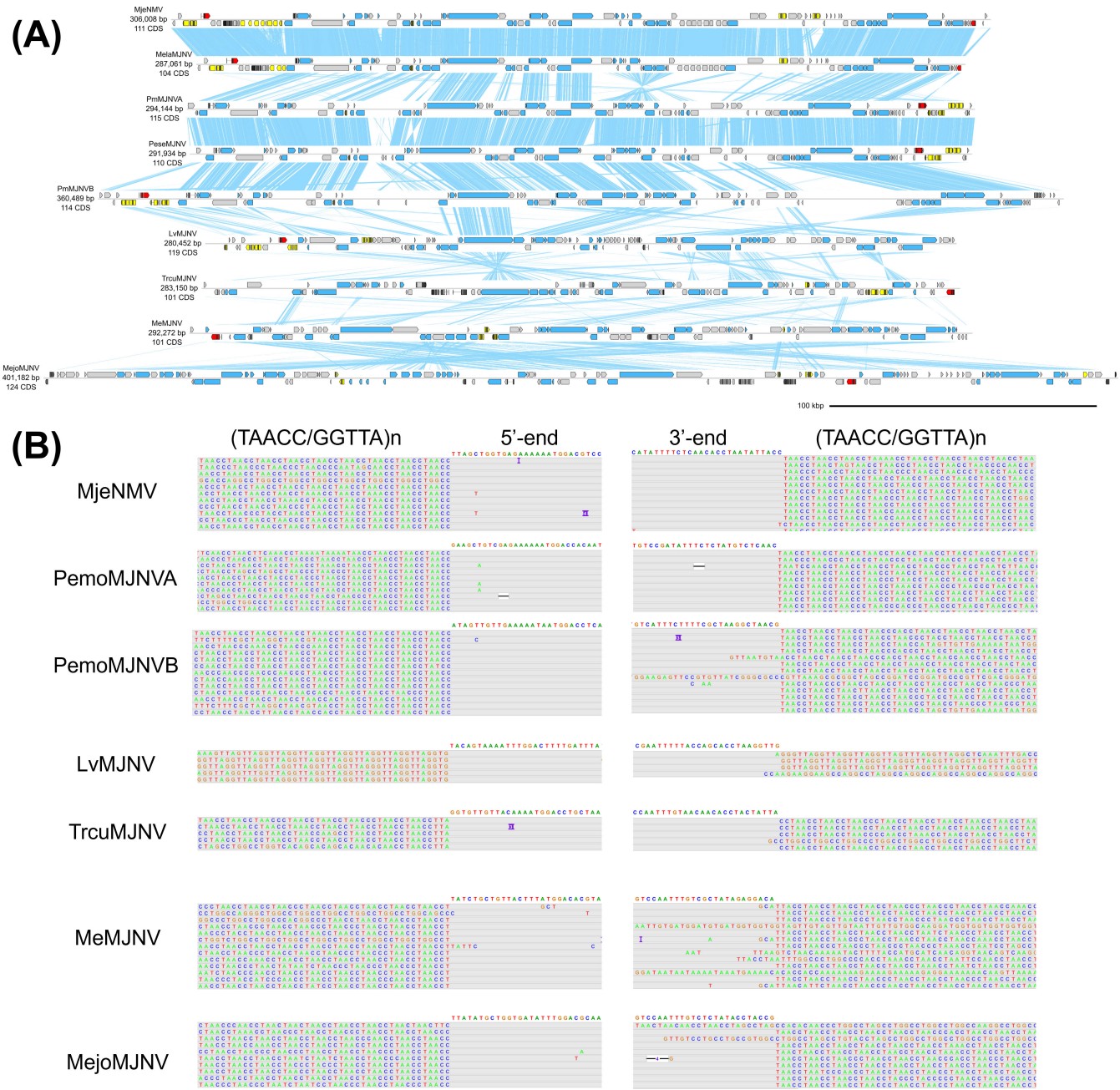

**FIG 2** Genome diagram of majaniviruses. (A) Genome diagrams of majaniviruses. Arrows indicate predicted genes and their transcriptional orientations; blue, WSSV homologs; yellow, baculoviral inhibitor of apoptosis repeat-containing proteins; gray, hypothetical and other eukaryotic-like proteins; red, tyrosine recombinases. Blue ribbons indicate pairwise TBLASTX hits (*e*-value :1−e3, bitscore: 50). (B) ONT read alignments flanking the 5′- and 3′-ends of majanivirus genomes.

## Pemoniviruses: another telomere-dwelling endogenous nimavirus lineage colonizing penaeid shrimp genomes

We reconstructed the genomes of Penaeus monodon endogenous nimavirus (PmeNMV; LC738869.1) and three related viruses, forming a novel genus-level clade which we have named Pemonivirus (**Pe**naeus **mo**nodon **ni**mavirus; Fig. 4A). Pemonivirus genomes range in size from 300 to 360 kb, with GC contents ranging from 43% to 48%. Pemoniviruses selectively insert into telomere motifs (Fig. 4B). Unlike majaniviruses, pemonivirus genomes contain few eukaryotic-derived genes. Pemoniviruses were found only from

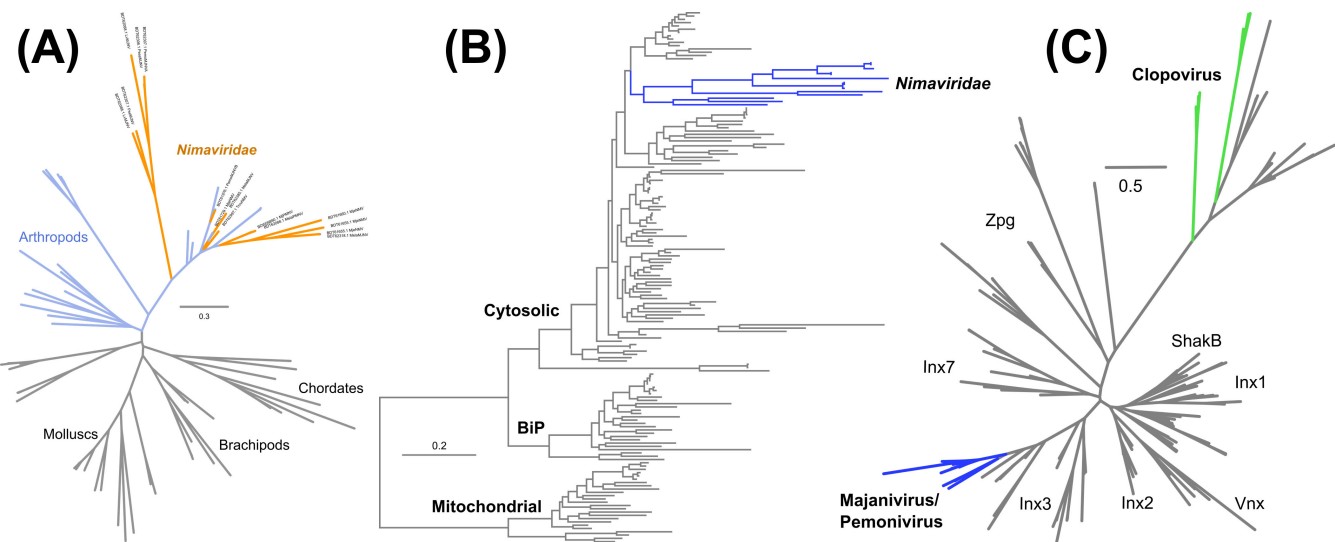

**FIG 3** Phylogenetic analysis of eukaryotic-like genes in majaniviruses. (A) Maximum-likelihood phylogenetic tree of 80 baculoviral inhihbitor of apoptosis repeat-containing proteins (266 sites; model: LG + I + G4). (B) Maximum-likelihood phylogenetic tree of 162 HSP70-like proteins (579 sites; model: LG + R7). (C) Maximum-likelihood phylogenetic tree of 158 innexin proteins (251 sites; model: LG + R6).

members of Penaeus *s. l.* occurring in the IWP region, including *Marsupenaeus japonicus*, *Melicertus latisulcatus*, *Penaeus semisulcatus*, and *Penaeus monodon*.

## Clopovirus: divergent, terrestrial isopod-associated nimaviruses

We identified a new clade of nimaviruses in the genomes of terrestrial isopods, which we named Clopovirus (derived from the French word for "pill bug," **clopo**rte) (Fig. 5). Clopovirus genomes range in size from 410 to 580 kb, making them the largest nimavirus genomes discovered to date. Armadillidium vulgare clopovirus (AvCLPV; LC738883.1) was assembled into a 416,069 bp sequence, coding for 120 protein-coding genes.

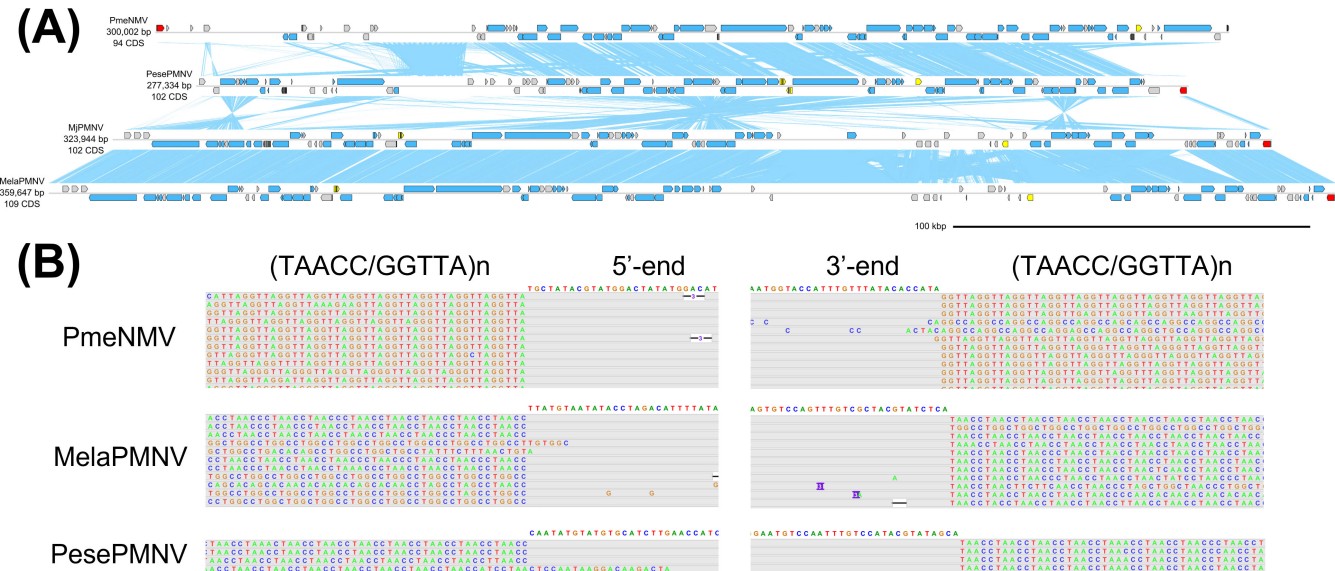

**FIG 4** Genome diagrams of pemoniviruses. (A) Genome diagrams of pemoniviruses. Arrows indicate predicted genes and their transcriptional orientations; blue, WSSV homologs; yellow, baculoviral inhibitor of apoptosis repeat-containing proteins; gray, hypothetical and other eukaryotic-like proteins; red, tyrosine recombinases. Blue ribbons indicate pairwise TBLASTX hits (e-value: 1−e3, bitscore: 50). (B) ONT read alignments flanking the 5′- and 3′-ends of pemonivirus genomes.

**TABLE 3** Nimaviral core genes found in clopovirus genomes

| Nimaviral core genes | | ID | AvCLPV | PsCLPV | TrCLPV |
|---|---|---|---|---|---|
| Structural proteins | Envelope proteins | wsv293a | | | |
| | | wsv327[a] | BDT63349.1 | BDV50114.1 | TRCLPV_1850 |
| | Capsid proteins | wsv037 | | | |
| | | wsv220 | | | |
| | | wsv271 | | | |
| | | wsv289 | BDT63373.1 | BDV50138.1 | TRCLPV_2110 |
| | | wsv308 | | | |
| | | wsv332 | | | |
| | | wsv360 | BDT63312.1 | BDV50077.1 | TRCLPV_0400 |
| | | wsv415 | | | |
| | Unknown | wsv131 | | | |
| | | wsv161 | | | |
| Nonstructural proteins | DNA polymerase | wsv514 | BDT63304.1 | BDV50068.1 | TRCLPV_0300 |
| | Helicase | wsv447 | BDT63392.1 | BDV49995.1 | TRCLPV_1500 |
| | AAA ATPase | wsv026 | BDT63390.1 | BDV49990.1 | TRCLPV_1520 |
| | Protein kinase | wsv423 | | | |
| | Latency-related gene | wsv427 | | | |
| | TATA-binding protein | wsv303 | BDT63332.1 | BDV50131.1 | TRCLPV_2040 |
| | Hypothetical protein | wsv137 | | | |
| | | wsv139 | | | |
| | | wsv192 | | | |
| | | wsv133 | | | |
| | | wsv267 | | | |
| | | wsv282 | BDT63296.1 | BDV50018.1 | TRCLPV_0140 |
| | | wsv285 | | | |
| | | wsv310 | | | |
| | | wsv313 | | | |
| | | wsv343 | BDT63282.1 | BDV50024.1 | TRCLPV_1200 |
| | | wsv433 | BDT63379.1 | BDV50142.1 | TRCLPV_1580 |
| | | wsv440 | | | |
| Missing in some nimaviruses | Envelope proteins | wsv011[a] | BDT63370.1 | BDV50135.1 | TRCLPV_2140 |
| | | wsv021 | BDT63369.1 | BDV50095.1 | TRCLPV_2140 |
| | | wsv035[a] | BDT63331.1 | BDV50132.1 | TRCLPV_2070 |
| | | wsv115[a] | BDT63386.1 | BDV49987.1 | TRCLPV_1560 |
| | | wsv209[a] | BDT63346.1 | BDV50116.1 | TRCLPV_1880 |
| | | wsv206 | | | |
| | | wsv216 | BDT63368.1 | BDV50096.1 | TRCLPV_0600 |
| | | wsv325 | BDT63338.1 | BDV50126.1 | TRCLPV_1970 |
| | | wsv432 | | | |
| | Tegument | wsv306[a] | BDT63353.1 | BDV50110.1 | TRCLPV_1810 |
| | Unknown | wsv134[a] | BDT63382.1 | BDV49996.1 | TRCLPV_1490 |
| | | wsv136 | | | |

[a]Naldaviral core genes.

AvCLPV might be the same virus as the WSSV-like sequences reported by Thézé et al. (25). Porcellio scaber clopovirus (PsCLPV; AP027154.1) was identified from the shotgun sequencing data of *Porcellio scaber*. The genome sequence of TrCLPV (File S3) was identified from the shotgun sequencing data of *Trachelipus rathkii*, a species native to Europe but introduced into North America (16). TrCLPV has a genome size of 579 kb, the largest of all nimaviruses discovered so far.

All clopovirus genomes contained a stretch of (TAACC/GGTTA)n repeats, suggesting that clopoviruses specifically insert into this sequence motif (Fig. 5B). However, most ONT reads mapping to these regions span the (TAACC/GGTTA)n repeat to align to either

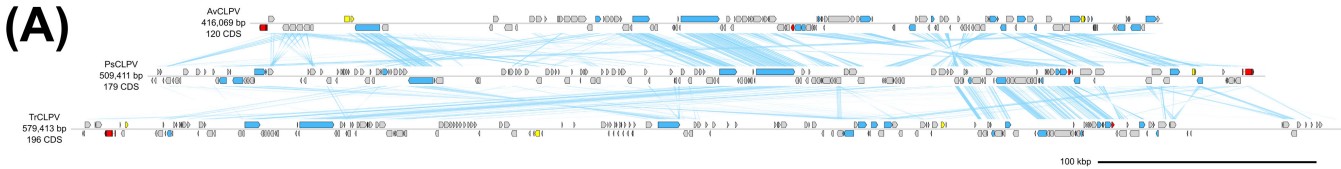

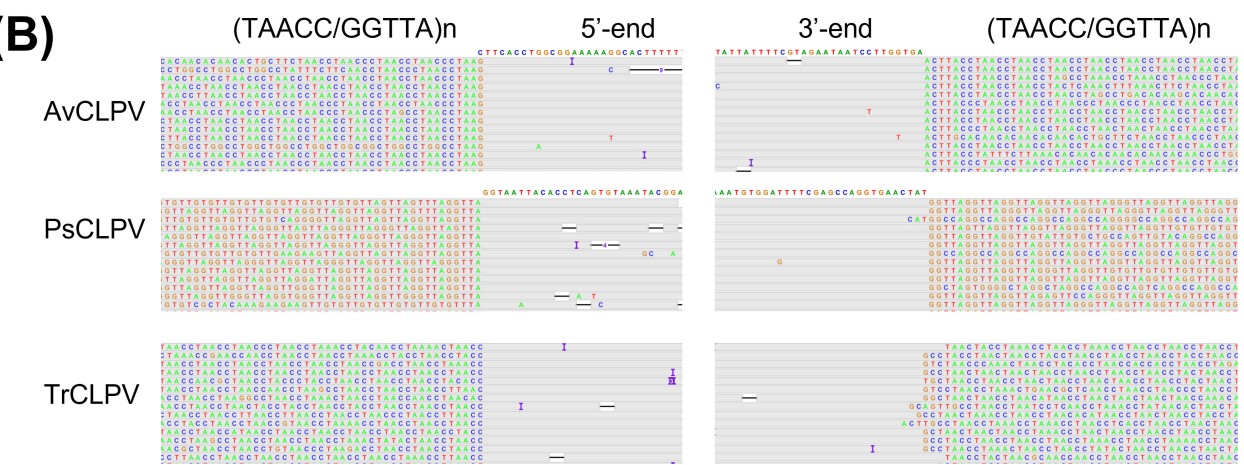

**FIG 5** The clopoviruses. (A) Genome diagrams of clopoviruses. Arrows indicate predicted genes and their transcriptional orientations; blue, WSSV homologs; yellow, baculoviral inhibitor of apoptosis repeat-containing proteins; gray, hypothetical and other eukaryotic-like proteins; red, tyrosine recombinases. Blue ribbons indicate pairwise TBLASTX hits (e-value: 1–e3, bitscore: 50). (B) ONT read alignments flanking the 5′- and 3′-ends of clopovirus genomes.

end of the clopovirus genome, suggesting that many of the clopovirus copies exist as episomes or concatemers. For consistency with other nimaviral MAGs, we removed (TAACC/GGTTA)n from the clopoviral genome assemblies to produce linear contigs.

Together, these results reveal the presence of a divergent nimavirus lineage in terrestrial isopods, which we call clopoviruses. Clopoviruses possessed 19 ancestral nimaviral genes, of which 10 are core genes (Table 3). Given the small number of genes shared with other nimaviruses, clopoviruses could be classified into a novel family.

## Whispoviruses

We believe that the remaining nimaviruses can be united under *Whispovirus*, the only genus currently recognized by ICTV, due to their coherent phylogenetic clustering (Fig. 1). Metapenaeus ensis nimavirus (MeNMV; LC738877.1), Trachysalambria curvirostris nimavirus (TrcuNMV; LC738880.1), and Sicyonia whispovirus (SicyWSV; LC738881.1) were flanked by (TAACC/GGTTA)n repeats (Fig. 6) and possessed tyrosine recombinases related to those found in majaniviruses and pemoniviruses (Fig. 7). Insertion specificities and associated integrases of other whispoviruses are discussed in the following sections.

## Telomere-associated tyrosine recombinases

All telomere-associated nimaviruses shared a distinct family of tyrosine recombinase (YR), a site-specific DNA recombinase that is pervasive in prokaryotes but is rarely documented in eukaryotes (26–28) (Fig. 7). The majaniviral YRs are arm-binding domain-containing tyrosine recombinases (29) that typically contained a C2H2 zinc-binding domain and a N-terminal SAM-like domain (Fig. 7A). We located five out of seven key residues important for YR functions (26, 30). The telomere-associated YRs are distinct from YRs encoded by nudiviruses (31) or NCLDV (32), nor are they closely related to known eukaryotic YR elements such as DIRS (33), Enterprise (34), and Cryptons (35, 36) (Fig. 7B and C).

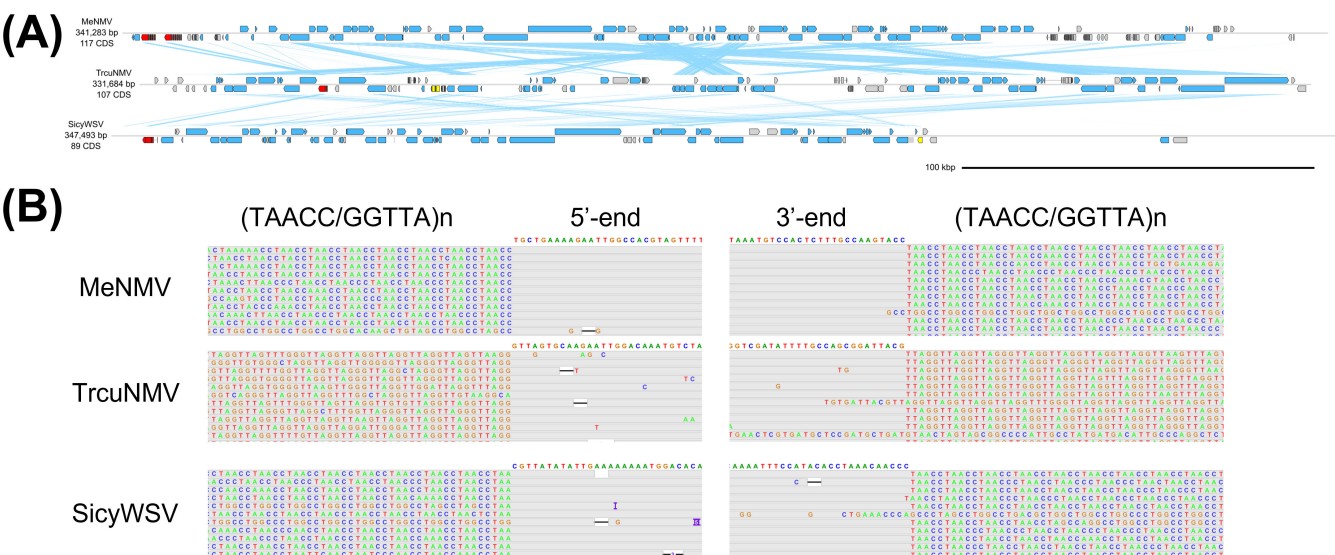

**FIG 6** Telomere-inserting whispoviruses. (A) Genome diagrams of whispoviruses. Arrows indicate predicted genes and their transcriptional orientations; blue, WSSV homologs; yellow, baculoviral inhibitor of apoptosis repeat-containing proteins; gray, hypothetical and other eukaryotic-like proteins; red, tyrosine recombinases. Blue ribbons indicate pairwise TBLASTX hits (*e*-value: 1–e3, bitscore: 50). (B) ONT read alignments flanking the 5′- and 3′-ends of whispovirus genomes.

## 28S rDNA-associated tyrosine recombinase in the closest WSSV relatives

PotrWSV (File S4) and Pasiphaea japonica whispovirus (PajaWSV; LC738885.1) are the closest relatives of WSSV analyzed in this study, forming a stem group leading to WSSV (Fig. 1 and 8). PotrWSV was discovered from the genome sequencing data of the swimming crab *Portunus trituberculatus* (14). PajaWSV, identified from the shotgun sequence data of the Japanese glass shrimp (*Pasiphaea japonica*), is the closest relative of WSSV characterized in this study. PajaWSV and WSSV share an average amino acid identity of 42.94%. PajaWSV and PotrWSV insert into the host 28S rDNA with a 11-mer target site duplication (5′-CCGTCGCGRGAC-3′), a conserved motif occurring within 28S rDNA (Fig. 8B and C).

PajaWSV and PotrWSV shared predicted multi-exon tyrosine recombinase genes that are phylogenetically related to telomere-associated YRs (Fig. 7). BLASTP search revealed additional YR-like proteins from decapod crustaceans although they were not associated with nimaviruses. Inclusion of the additional YRs into the phylogenetic tree led to the conclusion that PajaWSV and PotrWSV YRs were not immediate phylogenetic neighbors, raising the possibility that the YR genes in the two virus genomes were acquired independently. Collectively, these results suggest that two immediate WSSV relatives employ a distinct family of tyrosine recombinase to integrate into host 28S rDNA although whether the tyrosine recombinase genes are orthologous remains an open question.

## Ginger2 exaptation in sesarmid nimaviruses

Sesarmid crabs *Orisarma intermedium* (formerly known as *Sesarmops intermedium*) and *Orisarma dehaani* (formerly known as *Chiromantes dehaani*) harbor endogenous nimavirus genomes (Fig. 9) (10). The genome sequences of Sesarmops intermedium nimavirus (SiNMV; LC738884.1) and Chiromantes dehaani nimavirus (CdNMV; AP027155.1) were assembled into sequences of 265 kb and 285 kb, respectively. The two sesarmid nimavirus genomes are extensively colinear and share 94% average nucleotide identity. Both viruses insert into (AT/TA)n repeats and are flanked by a 46-base terminal inverted repeat (TIR: 5′-GTTGTGCCTAATAAGGATAATGACTCATTAACGCTAATAGGTAACG-3′). The presence of clearly defined inverted repeats was unique to the sesarmid nimaviruses.

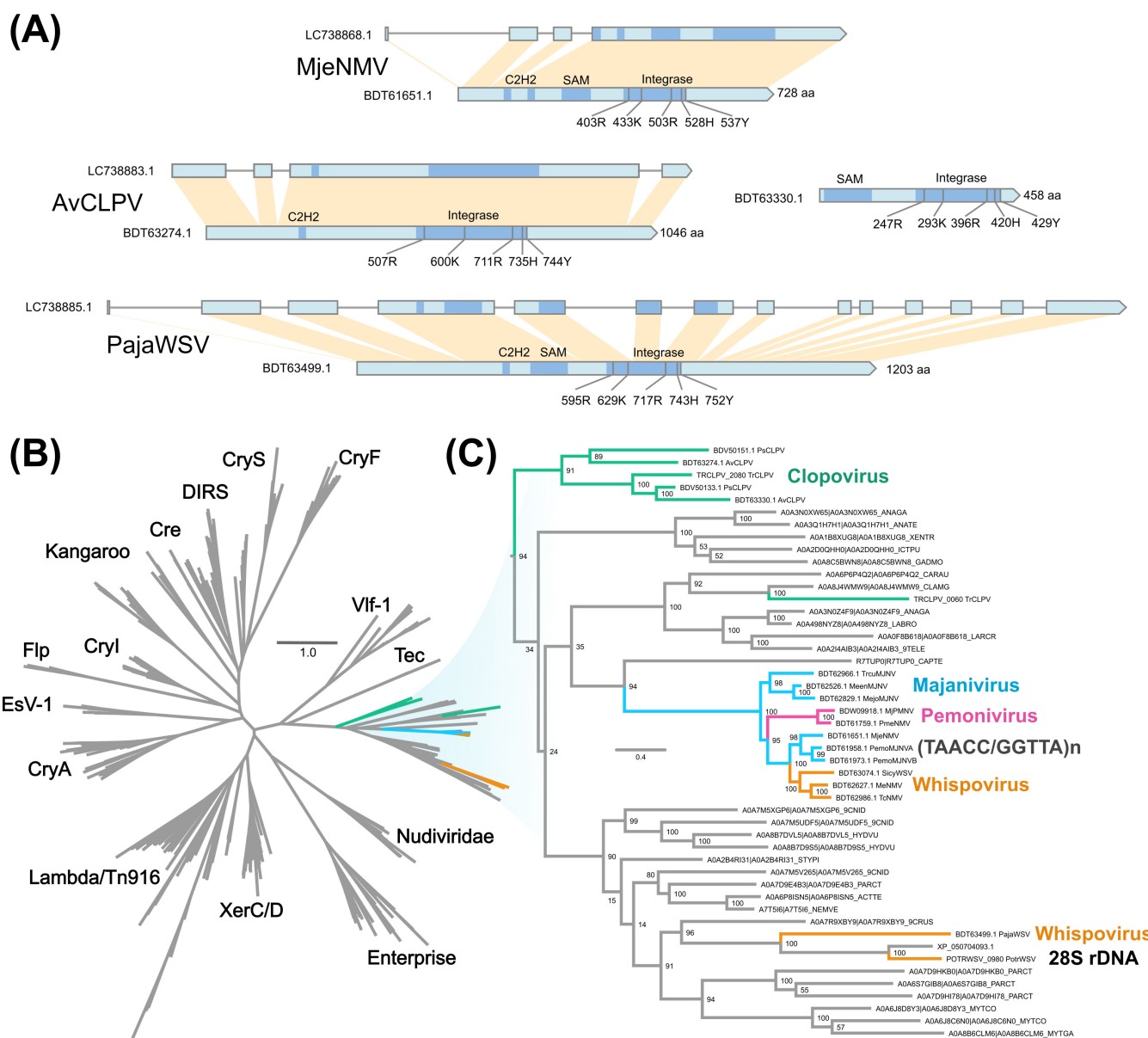

**FIG 7** Nimaviral tyrosine recombinases. (A) Gene diagram of nimaviral tyrosine recombinases. (B) Maximum-likelihood phylogenetic tree of 342 tyrosine recombinases (555 sites; model: LG + R7). (C) Subtree of (B) showing the phylogenetic relationships of the nimaviral tyrosine recombinases.

SiNMV and CdNMV genomes lacked YRs but possessed intron-containing genes with structural similarities to retroviral integrases (Fig. 9) (37). These nimaviral integrases clustered with Ginger2 transposable elements, a group of intron-containing cut-and-paste transposable element with TIRs (38). Hemigrapsus takanoi nimavirus (HtNMV; LC738882.1) also lacked YR and had a similar integrase gene (Fig. 9). Along with the presence of 46 bp TIR flanking the sesarmid nimavirus genomes, these results suggest that sesarmid endogenous nimaviruses, and possibly HtNMV, have adapted a distinct family of integrase-like genes for AT/TA-motif-specific integrations. This also suggests that sesarmid nimaviruses have a linear genome, as retroviral integrases act against linear templates (39).

## Limited expression of MjeNMV genes

To assess the transcriptional landscape of an endogenous nimavirus, we mapped multi-tissue RNA-seq data of *M. japonicus* (17) against the MjeNMV genome. The mapping rates

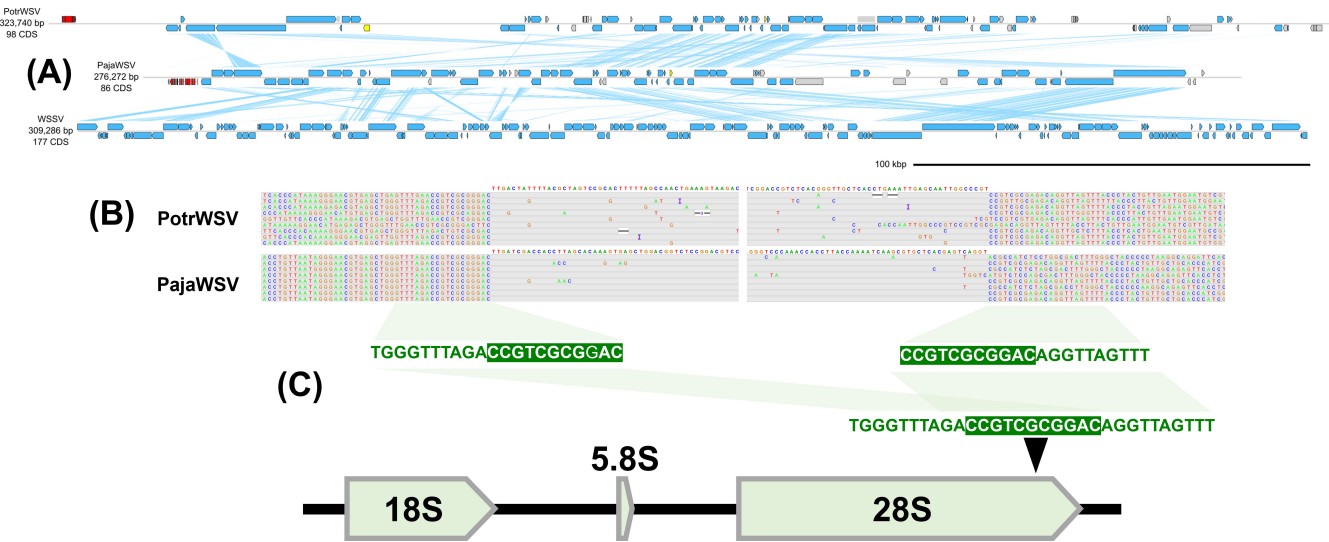

**FIG 8** Genome diagrams of 28S rDNA-specific whispoviruses. (A) Genome diagram of sesarmid nimavirus genomes. Arrows indicate predicted genes and their transcriptional orientations; blue, WSSV homologs; yellow, baculoviral inhibitor of apoptosis repeat-containing proteins; gray, hypothetical and other eukaryotic-like proteins; red, tyrosine recombinases. Blue ribbons indicate pairwise TBLASTX hits (e-value: 1−e3, bitscore: 50). (B) ONT read alignments flanking the 5′- and 3′-ends of 28S rDNA-specific whispoviruses. (C) 28S rDNA-specific whispovirus target site duplication.

were universally low (0.0003%–0.1114%), indicating that MjeNMV activity is limited to low level (Fig. S6). Regardless, their mapping profiles were strikingly different and deserved attention. MjeNMV transcripts in somatic tissues were predominantly short and antisense, whereas gonads had more sense-stranded transcripts. The presence of antisense transcripts in somatic tissues is suggestive of transcriptional silencing mediated by small RNAs such as siRNA and piRNA, while sense transcripts in gonads are suggestive of weak activity. Overall, transcriptional landscape of MjeNMV is evidently different between somatic and germline tissues although their functional significance remains unclear.

## Divergence time estimation of *Nimaviridae* using penaeid host phylogeography

Viruses do not leave fossil records, but we can trace their evolutionary history by linking it to that of their host (40–43). As described in a previous section, the evolutionary history of majaniviruses is tightly intertwined with the phylogeography of penaeid shrimps. This prompted us to use the shrimp phylogeography as a calibrator for estimating the evolutionary timelines of nimaviruses. *Penaeus s. l.* is considered to have originated in the present IWP and spread eastward to the AEP (44), where it diverged into two genera, *Litopenaeus* and *Farfantepenaeus*. Majaniviruses found from *Litopenaeus* and *Farfantepenaeus* were most likely introduced to AEP along with the *Litopenaeus-Farfantepenaeus* common ancestor. This assumption derives the divergence between the IWP and EAP to be as old as the divergence between *Litopenaeus* and *Farfantepenaeus*, which was estimated to be at least 42 million years ago (MYA) (Fig. S1 and S5).

LsMJNV and FdMJNV are found in the Eastern Pacific and the Atlantic, respectively. Gene flow between the two oceans has been shut off since the formation of the Isthmus of Panama, which occurred approximately 2.8 MYA (45) . This leads us to hypothesize that the divergence between LsMJNV and FdMJNV dates back to at least 2.8 MYA.

Using these two inferred calibration points, we estimated the divergence times of majaniviruses and the entire *Nimaviridae* family (Fig. 10). Overall, the divergence times of younger nodes appear more credible than older nodes. For example, the divergence between LdMJNV and FdMJNV was estimated to be 10.4 ± 1.2 MYA. The average nucleotide identity of the two viruses is 85% (15 substitutions/100 nucleotides), which

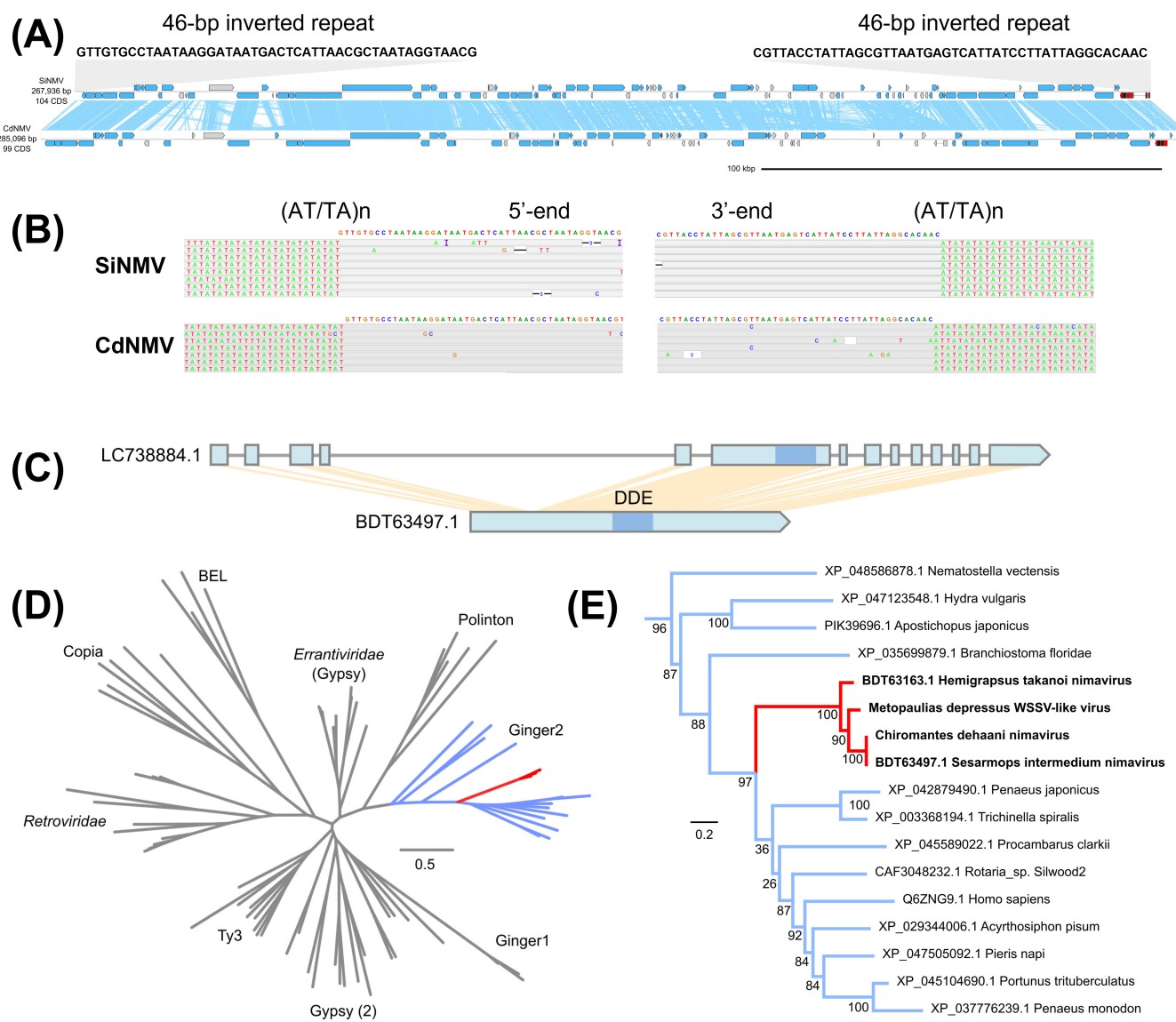

**FIG 9** Ginger2 exaptation in sesarmid nimaviruses. (A) Genome diagram of sesarmid nimavirus genomes. Arrows indicate predicted genes and their transcriptional orientations; blue, WSSV homologs; gray, hypothetical and eukaryotic-like proteins; red, integrases. Blue ribbons indicate pairwise TBLASTX hits (*e*-value: 1−e3, bitscore: 50). (B) 5′- and 3′-end flanking sequences of the SiNMV genome. (C) Gene diagram of the integrase-like gene in the SiNMV genome. (D) Maximum-likelihood phylogenetic tree of 57 DDE transposases (422 sites; model: Q.insect + I + G4). (E) Subtree of (D) showing the phylogenetic relationships of Ginger2-like elements.

translates to $1.44 \times 10^{-8}$ substitution/site/year, which falls within the range of known substitution rates of large double-stranded DNA viruses (46). Divergence time between SiNMV plus CiNMV and *Metopaulias depressus* WSSV-like virus was estimated to be 15.4 ± 1.8 MYA, which is way older than 4 MYA, the minimum estimated divergence time of *M. depresssus* and other sesarmid crabs (47). The crown Pemonivirus clade, whose members are exclusively found in IWP *Penaeus s. l.*, was estimated to be 25.4 ± 2.8 million years old. This is concordant with their absence from *Litpoenaeus and Farfantepenaeus*, which are estimated to have been isolated from the IWP for at least 42 million years.

The divergence dates of deep branches in the *Nimaviridae* family tree are younger than the estimated divergence of major host lineages. The last common ancestor of majaniviruses was estimated to be present 90 MYA, which is substantially younger than the early diversification of penaeid shrimps, which took place much earlier by the late Jurassic (48–51). Genus *Whispovirus* was estimated to date back to 171.9 ± 17.3 MYA. The

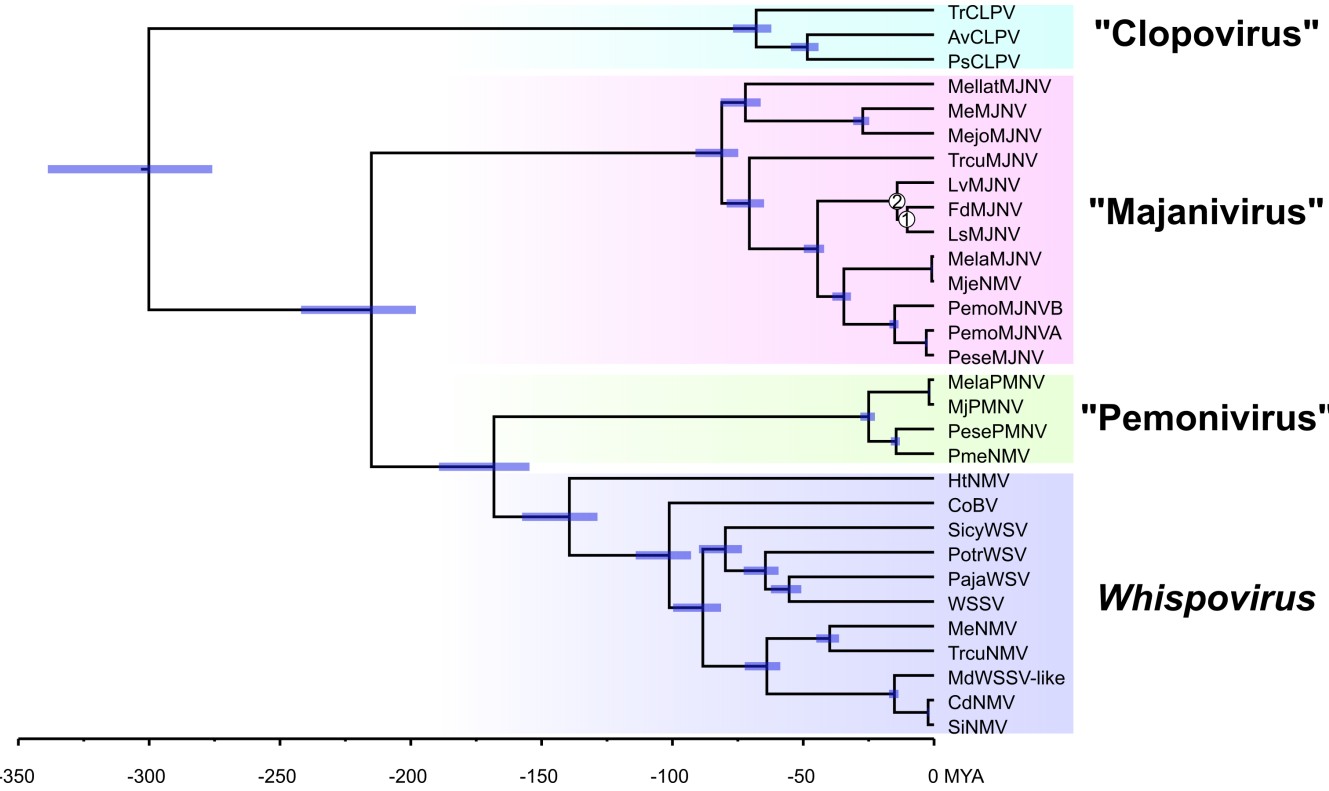

**FIG 10** Divergence time estimation of *Nimaviridae*. A total of nine nimaviral core protein sequences (12,905 sites) were used in the analysis. Blue bars indicate 95% confidence intervals of estimated divergence dates. Numbers on nodes correspond to two calibration points described in Table S5. All nodes were supported by a posterior probability of 1. Proposed viral genus names are quoted and unitalicized. WSSV, white spot syndrome virus; CoBV, Chionoecetes opilio bacilliform virus; MdWSSV-like, Metopaulias depressus WSSV-like virus; see Table 2 for the abbreviations for the other viruses.

divergence between isopod and decapod nimaviruses was estimated to be 307.3 ± 31.4 MYA, which is much younger than the estimated divergence time of eucarids and peracarids, which dates back to the Cambrian (52). These values suggest that majaniviral diversification is more characterized by jumping between closely related hosts than strict host-viral co-evolution.

Overall, the availability of multiple endogenous nimavirus genomes closely associated with a particular host taxon allows us to time-calibrate the evolutionary history of nimaviruses. The estimates support the idea that nimaviruses have been associated with crustacean hosts for the last few hundreds of millions of years.

## DISCUSSION

It has long been known that crustacean genomes harbor various WSSV-like sequences (7–9, 53), but the reasons why they are present has remained unknown. The present results demonstrate that endogenous nimaviruses selectively insert into specific genomic contexts, and this specificity is correlated with the types of integrases they encode. We propose that endogenous nimaviruses are selfish genetic elements that persist within the host genomes (54) and that the capture of integrase genes with different insertion specificities has allowed nimaviruses to persist as genomic parasites colonizing different repetitive motifs representing genomic niches (55, 56) . We note that these endogenous nimaviruses are distinct from fragmented viral insertions that may produce potentially immunogenic transcripts (57–59). The selfish nature of transposable elements could explain the persistence of endogenous nimaviruses even without a perceivable selective advantage to the host (60).

While it is possible that promiscuous integration followed by biased selective retention produced the appearance of selective integration, it is unlikely that only one of a wide variety of repetitive motifs present in the host genome would be tolerated. Therefore, the most likely explanation for the observed insertion selectivity is that it is mediated by site-specific integrases.

We now know that endogenous nimaviruses exist as multi-copy elements within host genomes. However, the process by which these populations formed remains unknown. One possibility is that a single ancestral infection event initiated a series of multiplications that gave rise to hundreds of copies within the host. Alternatively, closely related viruses—divergent at the strain or isolate level—might have repeatedly infected the same host, contributing to the increased copy numbers. The reality could be a combination of both scenarios, suggesting that endogenous nimaviruses may have experienced a complex evolutionary history. A detailed analysis of within-host sequence diversity could potentially allow us to infer the population dynamics of endogenous nimaviruses. However, we currently find this task to be challenging.

Overall, the distribution of integrases among nimaviruses does not strictly align with their phylogenetic relationships, indicating that nimaviruses have acquired integrases multiple times throughout evolutionary history. Interestingly, we observed that nimaviruses from phylogenetically distinct lineages, such as sesarmid nimaviruses and HtNMV, can possess mutually similar integrases, which raises the possibility that integrase genes may have been shared among different lineages of nimaviruses (61).

Integration of naldaviruses into host genomes has occurred many times throughout their evolutionary history (31, 62–66), and some, such as polydnaviruses, have even become domesticated to serve host functions (31, 67–69). Endogenization is also prevalent among herpesviruses, with the *Teratorn* element in medaka fish being a well-known example (70–73). Overall, it appears that nuclear double-stranded DNA viruses have evolved along a spectrum between endogenous and exogenous states, swaying between the two extremes.

Repeat-specific integration is believed to be a survival strategy employed by transposable elements in order to minimize negative effects on host fitness (74–76). The prevalence of telomere-repeat-specific nimaviruses in penaeid shrimps and terrestrial isopods may be due to a combination of this viral survival strategy and the abundance of simple sequence repeats, including telomere-like repeats (77), in the genomes of these organisms (16–18, 78, 79).

MjeNMV gene expression was universally and yet showed tissue-specific variations. The role of epigenetic factors in this process is highly probable and merits further exploration. The weak expression of MjeNMV genes in gonads suggests gonad-specific activation, a behavior that mirrors that of certain transposable elements. These elements, to ensure their survival, activate within the germline while maintaining dormancy in somatic tissues, thereby avoiding detrimental impacts on the host's fitness (80). It is plausible that endogenous nimaviruses may adopt a similar lifecycle. It would be ideal if we could analyze the differential expressions of integrants in different parts of the chromosomes, but unfortunately, it is extremely challenging because individual copies are mutually almost identical and are difficult to resolve.

Endogenous nimaviruses tenaciously retain structural protein genes, suggesting that they maintain the capability to form viral particles and transmit between hosts. We speculate that endogenous nimaviruses in crustacean genomes are analogous to prophages in bacterial genomes, which can remain dormant until certain stressors trigger their reactivation. Conservation of PIFs among endogenous nimaviruses is particularly noteworthy (21), as they may facilitate the oral transmission of viral particles through cannibalism of dead hosts, which is a common transmission route of WSSV (81–83).

We speculate that the absence of the integrase gene may significantly contribute to the evolution of a free-living, highly pathogenic nimavirus. WSSV and CoBV, the only isolated free-living nimaviruses to date, are entirely devoid of integrases. The absence of

an integrase implies a lack of ability for the virus to integrate itself into the host genome and propagate via vertical inheritance. Once the virus loses its vertical transmission capability, it is likely to become reliant on horizontal transmission for its survival. This shift in transmission strategy could foster the emergence of highly pathogenic variants.

Our analyses may be biased by incomplete taxon sampling of the host and the scarcity of exogenous nimavirus genomes. However, the lack of observed diversity could also reflect the actual rarity of exogenous nimaviruses circulating in the environment. To date, we have not been able to identify nimavirus-like sequences in environmental metagenomes, and despite the long history of modern shrimp aquaculture, WSSV remains the only pathogenic nimavirus of penaeid shrimps. We hypothesize that exogenous nimaviruses are rare and the emergence of a pathogenic nimavirus is an even more unusual event. To confirm this hypothesis, it would be valuable to conduct thorough metagenomic surveys, similar to one conducted in *Drosophila melanogaster* (84), to assess the prevalence of exogenous nimaviruses and other double-stranded DNA viruses in various crustacean species.

Estimating the ages of integration for individual nimaviruses in a given host genome is currently challenging due to the repetitive nature of viral copies and their integration sites. Nevertheless, we postulate that it is possible to infer the divergence times of nimaviruses at the species or genus level by associating them with the hosts' phylogeography. In this study, we aimed to estimate the divergence times of *Nimaviridae* using majaniviruses and their closely associated hosts, *Penaeus s. l.* By introducing two calibration points inferred from the phylogeography of *Penaeus s. l.*, we obtained divergence time estimates for *Nimaviridae* spanning hundreds of millions of years.

However, our analysis has clear limitations. Our divergence time estimates rely on only two inferred calibration points for majaniviruses and lack any for deeper nodes. Indeed, the estimated divergence dates for deeper nodes, such as the emergence of the *Whispovirus* genus and the split between decapod and isopod nimaviruses, appear to be younger than the estimated divergence times of major host lineages. It is possible that we may have significantly underestimated the true depths of these divergence times.

Looking forward, we believe that host phylogeography could prove to be a powerful tool in inferring the evolutionary history of nimaviruses. As we continue to accumulate crustacean genome data, we anticipate discovering additional endogenous nimavirus genomes, some of which may be highly host-specific. Such nimaviral lineages will be invaluable in calibrating viral evolutionary timelines based on host divergence.

In conclusion, the availability of endogenous nimavirus genomes provides unique opportunities for studying the diversity and evolution of crustacean-infecting large DNA viruses.

## MATERIALS AND METHODS

### General sequencing, assembly, and annotation strategy

Decapod crustacean genomes are gigabase-sized and extremely rich in repetitive elements, making whole-genome assembly challenging (7, 8, 17, 78, 85–87). To circumvent this difficulty, we performed shallow-depth genome survey sequencing and assembled the reads as a metagenome and picked up viral contigs. We also analyzed publicly available genomic sequences where available (14, 16).

### Crustacean genome survey sequencing

We sequenced a total of 17 crustacean genomes using Illumina and Oxford Nanopore Technologies (ONT) platforms. Genomic DNA was extracted from muscle or whole animal using phenol-chloroform-isoamyl alcohol extraction or MagAttract HMW DNA Kit (Qiagen) and further purified using NucleoBond columns (Machery Nagel). For some samples, genomic DNA was size-selected using a custom PEG/NaCl precipitation buffer [9% PEG8000 (wt/vol), 1 M NaCl, 10 mM Tris-HCl (pH 8.0)] (88). For four of the samples

(*Litopenaeus vannamei*, *Hemigrapsus takanoi*, *Sesarmops intermedium*, and *Chiromantes dehaani*), we used gDNA preparations generated in a previous study (10).

ONT long-read libraries were prepared using the Ligation Sequencing Kit (SQK-LSK109, ONT), NEBNext Companion Module for Oxford Nanopore Technologies Ligation Sequencing (E7180, NEB), and Agencourt AMPure XP beads (Beckman Coulter). Libraries were size selected using the PEG/NaCl precipitation buffer described above. The ONT libraries were sequenced on R9.4.1 flow cells, with multiple nuclease flush (EXP-WSH004, ONT) and priming (EXP-FLP002, ONT) before library loading. The fast5 files were base-called using Guppy v5.0.11, v5.0.13, or v5.0.16, with the super accuracy mode. The fast5 files of the previously published *M. japonius* ONT reads (Ginoza2017; BioSample Accession No. SAMD00276454) (17) were base-called using Guppy v5 v5.0.13 with super accuracy mode.

Library preparation and sequencing on Illumina HiSeq 4000 (2 × 150 bp) was carried out by Eurofins Genomics (Tokyo). Raw Illumina reads were quality trimmed by Fastp v0.20.1(89). The filtered Illumina reads were also used for *de novo* assembly and polishing.

## Publicly available data sets

Publicly available whole-genome shotgun sequence data of *Portunus trituberculatus*, *Litopenaeus stylirostris*, *Farfantepenaeus duorarum*, and *Trachelipus rhatkii* were downloaded from the NCBI database (Table S2). The raw reads were analyzed in a similar manner to other crustacean genome data.

## *De novo* assembly of shotgun sequence data and virus discovery

The filtered Illumina reads were *de novo* assembled using SPAdes (90). Combinations of SPAdes versions and parameters varied depending on the time of the analysis, sequencing coverage, and data complexity. The SPAdes assemblies were used to salvage low-copy nimaviral sequences that could not be fully recovered from ONT assemblies, including Nima-1_Lva and SiNMV.

The ONT reads were filtered by 5-, 10-, or 20-kb length cutoffs using SeqKit (91) and were *de novo* assembled by Flye v2.9 (92) in metagenome mode. The primary ONT assemblies were visualized by Bandage v0.8.1 (93) and screened for nimaviral sequences by TBLASTN searches querying WSSV proteins. The identified nimaviral contigs were used as the bait to map back the ONT reads by Minimap2 (94), and the mapped ONT reads were reassembled by Flye v2.9 (92) in normal mode or Canu v2.2 (95). This generated consensus nimaviral genome sequences that we believe are close representations of the original viral genomes.

The contigs were subjected to multiple rounds of polishing involving Medaka v1.4.3, HyPo v1.0.3 (96), Pilon v1.24 (97), and/or POLCA (98), using ONT reads and Illumina reads. The actual combinations of polishers differ between the viral genomes. ONT and Illumina reads were mapped backed by Minimap2 and visualized using IGV to inspect read coverage and misassemblies (99). Assembly errors were manually curated.

## Gene prediction and annotation

Endogenous nimaviruses contained eukaryotic-like genes with introns, which cannot be predicted by prokaryotic gene prediction programs. To recover both classes of protein-coding genes, we used different gene prediction programs and integrated the outputs into a nonredundant annotation. Open reading frames were predicted by Prodigal v2.6.3 (100), and the predicted proteins were queried against the proteomes of nimaviruses (WSSV, Marsupenaeus japonicus endogenous nimavirus, Penaeus monodon endogenous nimavirus, Hemigrapsus takanoi nimavirus, Metapenaeus ensis nimavirus, Sesarmops intermedium nimavirus; last accessed December 2021) and arthropods (*Marsupenaeus japonicus*, *Litopenaeus vannamei, Penaeus monodon*, *Portunus trituberculatus*, and *Homarus americanus*; last accessed December 2021) (101), using BLASTP.

The BLASTP output was merged by Automated Assignment of Human Readable Descriptions (AHRD) pipeline (102) into a table containing functional description. The genomic coordinates corresponding to nimaviral-like proteins were masked by BEDtools (103), and the remaining coordinates were forwarded to *ab initio* eukaryotic-like gene prediction by Augustus v3.3.3 (104) using the *Apis mellifera* gene model (11). The use of *Apis mellifera* gene model was inspired by Bao et al. (11). The predicted proteins [generated by gffread (105)] were BLASTP searched against the abovementioned nimaviral and arthropod proteomes, and the BLASTP output was passed to AHRD to generate final functional annotations. The GFF3 annotation files were converted into DDBJ flat files using GFF3toDDBJ (https://github.com/yamaton/gff3toddbj) and fFconv (https://www.ddbj.nig.ac.jp/ddbj/ume-e.html).

## Comparative genomic analysis and visualization

Genome diagrams were generated by a custom script (https://github.com/satoshika-wato/bio_small_scripts/blob/main/plot_linear_genome.py). Average nucleotide identity (ANI) and average amino acid identity (AAI) values were calculated by the ANI calculator (http://enve-omics.ce.gatech.edu/ani/) and the AAI calculator (http://enve-omics.ce.gatech.edu/aai/), respectively (106–108).

## Long read alignment

Length-filtered long reads were mapped onto the nimaviral genomes with Minimap2 with -Y (soft-clipping) option. The SAM alignments were processed with SAMtools (109) and visualized with Integrative Genomics Viewer v2.12.3 (99).

## Phylogenomic analysis

Amino acid sequences of nine nimaviral core genes (wsv026, wsv282, wsv289, wsv303, wsv343, wsv360, wsv433, wsv447, and wsv514) were aligned using MAFFT v7.505 (110), and the alignments were trimmed using trimAl v1.2 (111). Maximum likelihood phylogenetic analysis was performed using IQ-TREE2 v2.2.0.3 (112).

## Protein structural prediction

The known wsv308- or wsv310-like proteins and majaniviral counterparts were aligned separately with MAFFT. The alignments were used for structural prediction with ColabFold v1.3.0 (22, 23). Pairwise alignment scores were calculated on the DALI server (http://ekhidna2.biocenter.helsinki.fi/dali/) (24). The predicted protein structures were visualized with UCSF ChimeraX v1.4 (113).

## Analysis of integrase genes

Multiple sequence alignments of representative tyrosine recombinase families (*CryA*, *CryF*, *CryI*, *CryS*, *Kangaroo*, and *Tec*) were downloaded from Kojima et al. (36) and queried against UniRef30 2020 February version on the HHblits server (https://toolkit.tuebin-gen.mpg.de/tools/hhblits; last accessed September 13, 2022) (114). Cre recombinase, Enterprise, VLF1, XerCD, Tec, Tn916, and Lambda integrase homologs were prepared by querying individual proteins against UniRef30 (88) by HHblits or NCBI non-redundant protein database by BLASTP. The proteins were aligned on MAFFT server (last accessed 13 September 2022) (110) with default settings. The alignments were then iteratively refined using CD-HIT (75% identity cutoff) (115) and MaxAlign (116) implemented on the MAFFT server. The alignments were subjected to HHpred server to identify regions exhibiting similarities to YR domains, cropped, aligned by MAFFT, and further refined. The resulting YR entries were merged into a single alignment by MAFFT with the following options: --maxiterate 1000–-globalpair–-op 3.06–-ep 0.246. Phylogenetic analysis was conducted with IQ-TREE v2.2.0.3.

Protein sequences of Ginger2 and other DDE transposases were downloaded from the NCBI database and aligned by MAFFT, trimmed with trimAl, and phylogenetic analysis was conducted with IQ-TREE 2.2.0.3.

## Copy number estimation

Estimated copy numbers of endogenous nimavirus genomes were calculated as follow:

$$\text{Virus copy number} = \frac{\text{Virus sequencing depth}}{\text{Estimated genome coverage}}$$

$$\text{Estimated genome coverage} = \frac{\text{Total reads (bp)}}{\text{Estimated host genome size (bp)}}$$

Estimated host genome sizes were retrieved from literature (14, 17–19) and the Animal Genome Size Database (https://www.genomesize.com/).

## Penaeid mitochondrial genome assembly and annotation

Mitogenome sequences of *Metapenaeopsis lamellata* and *Sicyonia* sp. Kyushu2019 were characterized in this study. A contig representing the mitochondrial genome was extracted from a Flye assembly of >5 kb ONT reads. Trimmed Illumina reads were mapped onto the contig by minimap2 and iteratively polished by Pilon v1.24 (97). The mitogenome was annotated on the MITOS2 server (http://mitos2.bioinf.uni-leipzig.de/index.py) (117). The annotated mitochondrial genomes of the two species are available as Supplementary Files of the manuscript.

## Phylogenetic analysis and divergence time estimation of penaeid shrimps

A total of 32 mitogenome sequences derived from the suborder Dendrobranchiata, which encompasses penaeoid and sergestoid shrimps, were downloaded from the NCBI database (accessed June 2023; Table S5). The mitogenomes of *Sicyonia* sp. Kyushu2019 and *Metapenaeopsis lamellata* were generated in this study as described in the previous section. The predicted amino acid sequences of 13 protein-coding genes were aligned by MAFFT v7.520. The alignments were used for Bayesian phylogenetic analysis and divergence time estimation using BEAST v2.7.4 (118). A strict molecular clock, the WAG substitution model, and the Yule speciation model were selected. A total of five fossil and geological calibration points were included as described in Table S5 (45, 46, 119, 120). Ten-million iterations were performed, which were sampled every 10,000 steps after a 10% burn-in. We used Tracer v. 1.7.1 (121) to monitor the progress of the run and to ensure that the effective sampling sizes of all parameters were larger than 200. A maximum clade credibility tree was generated with TreeAnnotator (https://www.beast2.org/treeannotator/), which was visualized with FigTree v1.4 (http://tree.bio.ed.ac.uk/software/figtree/).

## Divergence time estimation of *Nimaviridae*

The multiple sequence alignments of nine nimviral core proteins used in the maximum likelihood phylogenetic analysis were used for the Bayesian phylogenetic analysis by BEAST v2.7.4. A strict molecular clock, the WAG substitution model, and the Yule speciation model were selected. Two calibration points were introduced as described in Table S5 (45). Ten-million iterations were performed, which were sampled every 10,000 steps after a 10% burn-in. We used Tracer v. 1.7.1 (121) to monitor the progress of the run and to ensure that the effective sampling sizes of all parameters were larger than 200. A maximum clade credibility tree was generated with TreeAnnotator and was visualized with FigTree v1.4.4.

## MjeNMV transcriptome analysis

A total of 49 *M. japonicus* RNA-seq data were downloaded from NCBI database (Table S4) (17). The raw Illumina reads were trimmed by Fastp v0.23.0, and the trimmed reads were mapped onto the MjeNMV genome by HISAT2 v2.2.1 (122). Mapped reads were separated according to the transcriptional orientation using SAMtools. The results were visualized using a custom script.

### ACKNOWLEDGMENTS

This research was supported by Grants-in-Aid for Scientific Research from the Japan Society for Promotion of Science (JSPS) (JSPS KAKENHI Grant Numbers JP15H02462, JP19H00949, and 19J21518) and by Science and Technology Research Partnership for Sustainable Development from the Japan Science and Technology Agency (SATREPS Grant Number JPMJSA1806).

### AUTHOR AFFILIATION

[1]Laboratory of Genome Science, Tokyo University of Marine Science and Technology, Tokyo, Japan

### AUTHOR ORCIDs

Satoshi Kawato  http://orcid.org/0000-0003-2401-5621
Hidehiro Kondo  http://orcid.org/0000-0001-5102-6831
Ikuo Hirono  http://orcid.org/0000-0002-2355-3121

### FUNDING

| Funder | Grant(s) | Author(s) |
| --- | --- | --- |
| MEXT | Japan Science and Technology Agency (JST) | JPMJSA1806 | Ikuo Hirono |
| MEXT | Japan Society for the Promotion of Science (JSPS) | 19J21518 | Satoshi Kawato |
| MEXT | Japan Society for the Promotion of Science (JSPS) | JP22H00379 | Ikuo Hirono |

### AUTHOR CONTRIBUTIONS

Satoshi Kawato, Conceptualization, Data curation, Formal analysis, Investigation, Methodology, Writing – original draft | Reiko Nozaki, Formal analysis, Investigation, Methodology | Hidehiro Kondo, Data curation, Writing – review and editing | Ikuo Hirono, Funding acquisition, Project administration, Resources, Supervision, Writing – review and editing

### DATA AVAILABILITY

The raw reads generated in this study are deposited to DDBJ/NCBI/ENA database under the BioProject ID PRJDB13888. The accession numbers of the nimaviral MAG assemblies are provided in Table 2. Colabfold predictions of wsv308 and wsv310 orthologs are available as Supplementary Files 1 and 2, respectively. TrCLPV MAG, protein sequences, and genome annotation are available as Supplementary File 3. PotrWSV MAG, protein sequences, and genome annotation are available as Supplementary Files 4. LsMJNV MAG, protein sequences, and genome annotation are available as Supplementary Files 5. FdMJNV MAG, protein sequences, and genome annotation are available as Supplementary Files 6. The mitochondrial genome sequence of Metapenaeopsis lamellata is available as Supplementary File 7. The mitochondrial genome sequence of Sicyonia sp. Fukuoka2019 is available as Supplementary File 8. Examples of codes used in this study are available as Supplementary File 9. Supplementary Files 1 to 9 are available on FigShare (https://doi.org/10.6084/m9.figshare.22012370.v1).

## ADDITIONAL FILES

The following material is available online.

## Supplemental Material

**Supplemental figures (Spectrum00559-23-s0001.docx).** Figures S1 to S6.
**Supplemental tables (Spectrum00559-23-s0002.xlsx).** Tables S1 to S5.

## Open Peer Review

**PEER REVIEW HISTORY (review-history.pdf).** An accounting of the reviewer comments and feedback.

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
