## [Reviewer comments · Microbiology Spectrum]

Microbiology Spectrum

Integrase-associated niche differentiation of endogenous large DNA viruses in crustaceans

Satoshi Kawato, Reiko Nozaki, Hidehiro Kondo, and Ikuo Hirono

Corresponding Author(s): Ikuo Hirono, Tokyo University of Marine Science and Technology

Review Timeline:

Submission Date:	February 7, 2023
Editorial Decision:	April 18, 2023
Revision Received:	July 10, 2023
Accepted:	November 15, 2023

Editor: Alison Sinclair

Reviewer(s): The reviewers have opted to remain anonymous.

Transaction Report:

DOI: <https://doi.org/10.1128/spectrum.00559-23>

April 18, 2023

Prof. Ikuo Hirono
Tokyo University of Marine Science and Technology
4-5-7, Konan
Minato-ku
Tokyo 108-8477
Japan

Re: Spectrum00559-23 (Integrase-associated niche differentiation of endogenous large DNA viruses in crustaceans)

Dear Prof. Ikuo Hirono:

Thank you for submitting your manuscript to Microbiology Spectrum.

Please can you respond to each of the reviewers' comments.

Link Not Available

Sincerely,

Alison Sinclair

Journals Department
Reviewer comments:

Reviewer #1 (Comments for the Author):

In this manuscript, the author studied the genomes of endogenous nimaviruses collected from various sources and reports following novel findings.

1. Based on phylogenetic analysis, three novel genera of endogenous nimaviruses, which are named Majanivirus, Pemonivirus and Clopovirus, are proposed.
2. Endogenous nimaviruses are specifically inserted into arthropod telomere repeat motif or 28s rDNA and the insertion might be

associated with virus-encoded tyrosine recombinases or transposase.
There are some comments regarding results and description in the article.

Results

1. Description of "Table 1" was not included in this section.
2. In line 81-82 "The genomes of *Portunus trituberculatus* whispovirus and *Trachelipus rathkii* clopovirus are available as supplementary files of the manuscript."
To prevent confusion, it is recommended to specify the location of the supplementary files in the manuscript. Author should include figshare link.
3. Is there any difference between (TAACC/GGTTA)_n motifs in line 103 and (GGTTA/TAACC)_n motifs in line 106 ?
4. Line 107 to Line 110. "However, some ONT reads from one end of the (GGTTA/TAACC)_n tract to the other end of the majaniviral genomes could be aligned, suggesting that some majaniviral copies are present as concatemers and/or episomes."
Please rephrase the sentence to better explain the results.
5. Line 121, Please change Figure 3 to Figure 3B and C for better clarification.

Discussion

1. Line 285-287. "We hypothesize that exogenous nimaviruses are rare and the emergence of a pathogenic nimavirus is an even more unusual event."
What could be the reason for this phenomenon? Are there any possible limiting factors preventing the re-activation of endogenous nimavirus within the host? I believe that answering these questions could shed light on why WSSV is the only pathogenic nimavirus that has been observed.
2. As the nimavirus genome resided in the host genome, is it possible that some of the viral gene is expressed without the viral re-activation?

Reviewer #2 (Comments for the Author):

Kawato et al. report on the presence and classification of endogenous large DNA viruses (nimaviruses) that colonized crustacean genomes. By utilizing sequences in existing databases and sequencing other crustacean genomes isolated from species circulating in Japan, the authors assembled 25 endogenous nimaviral genomes. Phylogenetic analysis of the nimaviral core proteins reveal four major clusters. The authors provide descriptive analysis of these major clusters and highlight their unique features, including presence of eukaryotic-like genes. Integration-site analysis also reveals differences in integration sites possibly driven by the utilization of different recombinase/integrase enzymes. Overall, while the study is largely descriptive, it is nevertheless interesting and provides a unique resource for analysis of endogenized DNA viruses. I have the following questions/comments that can enhance the impact of the study.

- 1- The authors provide consensus sequences of the nimaviral genomes that are present in repetitive regions. Is there any estimate on how many integrants are present per genome analyzed? Likewise, can the authors provide an estimate on how many times nimaviruses may have integrated vs. duplicated following integration?
- 2- Can the authors provide an estimate on when different classes of nimaviruses may have integrated in the respective species they were isolated from?
- 3- Retroviral integrases work on linear dsDNA. It would be important to highlight whether nimaviruses genomes are linear or circular. If circular, the authors should discuss whether nimaviral recombinases/integrases can work on the genome template.
- 4- Are the newly identified nimaviral genomes all transcriptionally silent? It would be interesting to see whether different integrations sites dictates transcriptional activity.
- 5- Minor: Page 6-title: colonize to colonize
- 6- Minor: Page 7-line 126: core genes are a set "of" genes

Staff Comments:

Preparing Revision Guidelines

- Point-by-point responses to the issues raised by the reviewers in a file named "Response to Reviewers," NOT IN YOUR COVER LETTER.
- Upload a compare copy of the manuscript (without figures) as a "Marked-Up Manuscript" file.

- Each figure must be uploaded as a separate file, and any multipanel figures must be assembled into one file.
- Manuscript: A .DOC version of the revised manuscript
- Figures: Editable, high-resolution, individual figure files are required at revision, TIFF or EPS files are preferred

Please return the manuscript within 60 days; if you cannot complete the modification within this time period, please contact me. If you do not wish to modify the manuscript and prefer to submit it to another journal, please notify me of your decision immediately so that the manuscript may be formally withdrawn from consideration by Microbiology Spectrum.

Kawato et al. report on the presence and classification of endogenous large DNA viruses (nimaviruses) that colonized crustacean genomes. By utilizing sequences in existing databases and sequencing other crustacean genomes isolated from species circulating in Japan, the authors assembled 25 endogenous nimaviral genomes. Phylogenetic analysis of the nimaviral core proteins reveal four major clusters. The authors provide descriptive analysis of these major clusters and highlight their unique features, including presence of eukaryotic-like genes. Integration-site analysis also reveals differences in integration sites possibly driven by the utilization of different recombinase/integrase enzymes. Overall, while the study is largely descriptive, it is nevertheless interesting and provides a unique resource for analysis of endogenized DNA viruses. I have the following questions/comments that can enhance the impact of the study.

- 1- The authors provide consensus sequences of the nimaviral genomes that are present in repetitive regions. Is there any estimate on how many integrants are present per genome analyzed? Likewise, can the authors provide an estimate on how many times nimaviruses may have integrated vs. duplicated following integration?
- 2- Can the authors provide an estimate on when different classes of nimaviruses may have integrated in the respective species they were isolated from?
- 3- Retroviral integrases work on linear dsDNA. It would be important to highlight whether nimaviruses genomes are linear or circular. If circular, the authors should discuss whether nimaviral recombinases/integrases can work on the genome template.
- 4- Are the newly identified nimaviral genomes all transcriptionally silent? It would be interesting to see whether different integrations sites dictates transcriptional activity.
- 5- Minor: Page 6-title: colonize→colonize
- 6- Minor: Page 7-line 126: core genes are a set of genes

We would like to thank the reviewers for careful and thorough reading of this manuscript and for the thoughtful comments and constructive suggestions.

To reflect reviewers' insights in the manuscript, we have introduced two additional paragraphs in the Results section:

“Limited expression of MjeNMV genes” (lines 264-275): We analyzed the transcriptome of *Marsupenaeus japonicus* endogenous nimavirus (MjeNMV) in various tissues of the kuruma shrimp *Marsupenaeus japonicus*.

“Divergence time estimation of *Nimaviridae* using penaeid host phylogeography” (lines 276-321): We attempted divergence time estimation of *Nimaviridae* using time calibration points inferred from the phylogeography of *Penaeus sensu lato*, the host of majaniviruses.

Line numbers in the Responses refer to those in the revised manuscript.

Response to Reviewer 1:

Thank you for your careful review of the manuscript. Our response follows in a point-to-point manner below.

Comment:

Results

1. Description of "Table 1" was not included in this section.

Response:

The text has been modified accordingly (line 74).

Comment:

2. In line 81-82 "The genomes of *Portunus trituberculatus* whispovirus and *Trachelipus rathkii* clopovirus are available as supplementary files of the manuscript."

To prevent confusion, it is recommended to specify the location of the supplementary files in the manuscript. Author should include figshare link.

Response:

The text has been revised to guide the reader to the FigShare repository.

Lines 83-87: "The genomes of *Portunus trituberculatus* whispovirus (PotrWSV), *Litopenaeus stylirostris* majanivirus (LsMJNV), *Farfantepenaeus duorarum* majanivirus (FdMJNV), and *Trachelipus rathkii* clopovirus (TrCLPV) are available as supplementary files of the manuscript (see "Data availability" for the link to the FigShare repository)."

Lines 609-619: "TrCLPV MAG, protein sequences, and genome annotation are available as Supplementary File 3. PotrWSV MAG, protein sequences, and genome annotation are available as Supplementary Files 4. LsMJNV MAG, protein sequences, and genome annotation are available as Supplementary Files 5. FdMJNV MAG, protein sequences, and genome annotation are available as Supplementary Files 6. The mitochondrial genome sequence of *Metapenaeopsis lamellata* is available as Supplementary File 7. The mitochondrial genome sequence of *Sicyonia* sp. Fukuoka2019 is available as Supplementary File 8. Examples of codes used in this study are available as Supplementary File 9. Supplementary Files 1-9 are available on FigShare (<https://figshare.com/s/8029ba00a880cf00a7f0>)."

Comment:

3. Is there any difference between (TAACC/GGTTA)_n motifs in line 103 and (GGTTA/TAACC)_n motifs in line 106 ?

Response:

There is no difference. The text has been corrected to "(TAACC/GGTTA)_n".

Comment:

4. Line 107 to Line 110. "However, some ONT reads from one end of the (GGTTA/TAACC)_n tract to the other end of the majaniviral genomes could be aligned, suggesting that some majaniviral copies are present as concatemers and/or episomes."

Please rephrase the sentence to better explain the results.

Response:

Sorry for the confusion. We hope the modified text better describe the findings.

Lines 128-132: "However, some ONT reads were successfully mapped from one end of the majanivirus genome, spanning across the external (TAACC/GGTTA)_n tract, and reaching to the other end of the genome. This suggests that some majaniviral copies could exist as concatemers, episomes, or possibly as a combination of both."

Comment:

5. Line 121, Please change Figure 3 to Figure 3B and C for better clarification.

Response:

The text has been modified accordingly.

Lines 143-145: "...vertically inherited from a common ancestor of the majaniviruses (**Figure 3B and 3C**). BIR-containing proteins clustered with other decapod proteins, but we surmise that they are nimaviral sequences annotated as host genes (**Figure 3A**)."

Comment:

Discussion

1. Line 285-287. "We hypothesize that exogenous nimaviruses are rare and the emergence of a pathogenic nimavirus is an even more unusual event."

What could be the reason for this phenomenon? Are there any possible limiting factors preventing the re-activation of endogenous nimavirus within the host? I believe that answering these questions could shed light on why WSSV is the only pathogenic nimavirus that has been observed.

Response:

Thank you for raising an important point. Unfortunately, we don't have a clear explanation as to why endogenous nimaviruses discovered in crustacean genomes do not readily reactivate. Epigenetic and post-transcriptional factors are likely at play. In the kuruma shrimp, the endogenous nimavirus appears to be selectively transcribed in the ovary and testis, suggesting a germline-specific lifestyle analogous to those observed in some transposable elements. Furthermore, the absence of integrases in the exogenous nimaviruses (although only two have been discovered so far) may provide a vital clue to understanding the emergence of a pathogenic nimavirus.

In the revised manuscript, we have added the following passage:

Lines 388-395: “We speculate that the absence of the integrase gene may significantly contribute to the evolution of a free-living, highly pathogenic nimavirus. WSSV and CoBV, the only isolated free-living nimaviruses to date, are entirely devoid of integrases. The absence of an integrase implies a lack of ability for the virus to integrate itself into the host genome and propagate via vertical inheritance. Once the virus loses its vertical transmission capability, it is likely to become reliant on horizontal transmission for its survival. This shift in transmission strategy could foster the emergence of highly pathogenic variants.”

Comment:

2. As the nimavirus genome resided in the host genome, is it possible that some of the viral gene is expressed without the viral re-activation?

Response:

We investigated this point by mapping RNA-seq data against the MjeNMV genome. Nimaviral gene expression is quite limited, yet it varies among different tissues. Sense-strand transcripts appear to be primarily expressed in the gonads. We've addressed these findings in a new section of the Results (“Limited expression of MjeNMV genes”; lines 264-275) and also included additional discussion.

Lines 258-269:

“Limited expression of MjeNMV genes

To assess the transcriptional landscape of an endogenous nimavirus, we mapped multi-tissue RNA-seq data of *M. japonicus* (17) against the MjeNMV genome. The mapping rates were universally low (0.0003% to 0.1114%), indicating that MjeNMV activity is limited to low level (Supplementary Figure 6). Regardless, their mapping profiles were strikingly different and deserved attention. MjeNMV transcripts in somatic tissues were predominantly short and antisense, whereas gonads had more sense-stranded transcripts. The presence of antisense transcripts in somatic tissues is suggestive of transcriptional silencing mediated by small RNAs such as siRNA and piRNA, while sense transcripts in gonads are suggestive of weak activity. Overall, transcriptional landscape of MjeNMV is evidently different between somatic and germline tissues, although their functional significance remains unclear.”

Lines 370-379: “MjeNMV gene expression was universally and yet showed tissue-specific variations. The role of epigenetic factors in this process is highly probable and merits further exploration. The weak expression of endogenous nimaviruses in gonads suggests gonad-specific activation, a behavior that mirrors that of certain transposable elements. These elements, to ensure their survival, activate within the germline while maintaining dormancy in somatic tissues, thereby avoiding detrimental impacts on the host's fitness (79). It is plausible that endogenous nimaviruses may adopt a similar lifecycle. It would be ideal if we could analyze the differential expressions of

integrants in different parts of the chromosomes, but unfortunately it is extremely challenging because individual copies are mutually almost identical and are difficult to resolve.”

Response to Reviewer 2:

Thank you for your careful review of the manuscript. Our response follows in a point-to-point manner below.

Comment:

1. The authors provide consensus sequences of the nimaviral genomes that are present in repetitive regions. Is there any estimate on how many integrants are present per genome analyzed?

Response:

We have estimated the copy numbers of endogenous nimaviruses based on sequencing coverage, where genome size estimates for the host are available.

Lines 93-98: “We estimated the copy numbers of endogenous nimaviruses by calculating the genome sequencing coverage of the host from the volume of Illumina read data plus the estimated host genome sizes(14, 17–19) . The estimated copy numbers per haploid genome ranged from 22 (PotrWSV) to 477 (Marsupenaesus japonicus endogenous nimavirus; MjeNMV; LC738868.1) (Supplementary Table 3). The abundance of MjeNMV copies in the kuruma shrimp genome aligns with previous estimates (7, 20).”

Comment:

Likewise, can the authors provide an estimate on how many times nimaviruses may have integrated vs. duplicated following integration?

Response:

Unfortunately we were not able to estimate the number of genome invasion events in each host. Admittedly, our consensus MAGs are a lossy compression of the actual diversity of endogenous nimaviruses. We speculate it could be possible to infer the population dynamics of endogenous nimaviruses through the analysis of sequence diversity in a given host, but we could not come up with a way to actually do so. We have noted this in the Discussion.

Lines 341-350: “We now know that endogenous nimaviruses exist as multi-copy elements within host genomes. However, the process by which these populations formed remains unknown. One possibility is that a single ancestral infection event initiated a series of multiplications that gave rise to hundreds of copies within the host. Alternatively, closely related viruses—divergent at the strain or isolate level—might have repeatedly infected the same host, contributing to the increased copy numbers. The reality could be a combination of both scenarios, suggesting that endogenous nimaviruses may have experienced a complex evolutionary history. A detailed analysis of within-host sequence diversity could potentially allow us to infer the population dynamics of endogenous nimaviruses. However, we currently find this task to be challenging.”

Comment:

1- Can the authors provide an estimate on when different classes of nimaviruses may have integrated in the respective species they were isolated from?

Response:

Thank you for highlighting an important point. At present, we find it challenging to estimate the integration times for each species. This task would necessitate resolving individual loci, verifying vertical inheritance, sequencing the host at the population level, and sequencing sibling species, among other requirements.

We understand the essence of your suggestion to be the incorporation of a temporal axis in the analysis of *Nimaviridae* evolution. Indeed, we think it is feasible to infer evolutionary timelines by capitalizing on the close association of these elements with their host. This assumption does not mandate strict vertical inheritance. We can estimate the duration of association between these viral lineages and their hosts, irrespective of whether they are strictly coevolving or circulating among closely related hosts.

To support this effort, we have characterized two additional majaniviruses. These representatives are found on either side of the Isthmus of Panama, which formed approximately 2.8 million years ago. We also inferred another calibration point based on the phylogeography of *Panaeus sensu lato*, which radiated from the Indo-Western Pacific and the Atlantic-Eastern Pacific. We used these two calibration points to estimate the divergence times of the *Nimaviridae* family. We have added a new section in the Results (“Divergence time estimation of *Nimaviridae* using penaeid host phylogeography”; lines 276-321, Figure 10, lines 687-693)

Comment:

2- Retroviral integrases work on linear dsDNA. It would be important to highlight whether nimaviruses genomes are linear or circular. If circular, the authors should discuss whether nimaviral recombinases/integrases can work on the genome template.

Response

Thank you for pointing this out. Given the presence of TIR, sesarimid nimavirus genomes could be linear and employ a mechanism similar to that of retroviruses. We have inserted the following passage in the revised manuscript:

Lines 262-263: “This also suggests that sesarimid nimaviruses have a linear genome, as retroviral integrases act against linear templates (39).”

Comment:

3- Are the newly identified nimaviral genomes all transcriptionally silent? It would be interesting to see whether different integration sites dictate transcriptional activity.

Response

We investigated this by mapping RNA-seq reads from the kuruma shrimp against the MjeNMV genome ("Limited expression of MjeNMV genes"; Lines 264-275). While MjeNMV genes are mostly silent, they do appear to be expressed in the ovary and testis. Unfortunately, at present, it's challenging to attribute the transcripts to individual viral copies.

Regarding the impact of integration sites on copy-specific transcriptional activity, we might need a very high-quality, chromosome-level assembly generated from HiFi reads to answer this question. Such an assembly would be able to resolve copy-specific SNPs across nimaviral copies in each organism. We are attempting to generate such an assembly in the kuruma shrimp, but obtaining HiFi reads in this species has proven to be very challenging.

Comment:

4- Minor: Page 6-title: colonize to colonize

5- Minor: Page 7-line 126: core genes are a set "of" genes

Response

The text has been revised accordingly.

Re: Spectrum00559-23R1 (Integrase-associated niche differentiation of endogenous large DNA viruses in crustaceans)

Dear Prof. Ikuo Hirono:

Your manuscript has been accepted, and I am forwarding it to the ASM production staff for publication. Your paper will first be checked to make sure all elements meet the technical requirements. ASM staff will contact you if anything needs to be revised before copyediting and production can begin. Otherwise, you will be notified when your proofs are ready to be viewed.

Sincerely,
Clinton Jones
Editor
Microbiology Spectrum

Reviewer #1 (Comments for the Author):

non

Reviewer #2 (Comments for the Author):

My comments have been sufficiently addressed and I am aware of the reasons why certain analyses/predictions could not be done. Overall, I think this is a very interesting study and is well written.